# Transition to naïve human pluripotency mirrors pan-cancer DNA hypermethylation

Hemalvi Patani [1], Michael D. Rushton[1], Jonathan Higham[2], Saul A. Teijeiro[1], David Oxley[3], Pedro Cutillas [1], Duncan Sproul [2] & Gabriella Ficz [1✉]

Epigenetic reprogramming is a cancer hallmark, but how it unfolds during early neoplastic events and its role in carcinogenesis and cancer progression is not fully understood. Here we show that resetting from primed to naïve human pluripotency results in acquisition of a DNA methylation landscape mirroring the cancer DNA methylome, with gradual hypermethylation of bivalent developmental genes. We identify a dichotomy between bivalent genes that do and do not become hypermethylated, which is also mirrored in cancer. We find that loss of H3K4me3 at bivalent regions is associated with gain of methylation. Additionally, we observe that promoter CpG island hypermethylation is not restricted solely to emerging naïve cells, suggesting that it is a feature of a heterogeneous intermediate population during resetting. These results indicate that transition to naïve pluripotency and oncogenic transformation share common epigenetic trajectories, which implicates reprogramming and the pluripotency network as a central hub in cancer formation.

[1] Centre for Haemato-Oncology, Barts Cancer Institute, Queen Mary University of London, EC1M 6BQ London, UK. [2] MRC Human Genetics Unit and Edinburgh Cancer Research Centre, MRC Institute of Genetics & Molecular Medicine, University of Edinburgh, Western General Hospital, Crewe Road, EH4 2XU Edinburgh, UK. [3] Mass Spectrometry Facility, Babraham Institute, CB22 3AT Cambridge, UK. ✉email: g.ficz@qmul.ac.uk

D isruption of DNA methylation patterns is a hallmark of human cancers, typically characterised by loss of global genomic DNA methylation accompanied by site-specific hypermethylation[1–4]. DNA hypomethylation is typically associated with genomic instability[5,6], while site-specific DNA hypermethylation occurs at promoter CpG islands (CGIs) and can be associated with repression of tumour suppressor genes in cancer cells[7,8]. The majority of these observations have been made in cancer cell lines or primary cancer cells, but they are not fully representative of the processes occurring during the transition of normal cells into malignant cells. The underlying mechanisms that give rise to these opposing patterns of genomic DNA methylation in early stages of human cancer development remain elusive, as does the timing and biological function of such events in cancer initiation and progression. To this end, a recent study demonstrated that ageing and cancer associated DNA hypermethylation accelerates cellular transformation in a $Braf^{V600E}$ mouse colon organoid system, through suppression of Wnt signalling regulators in a progressive manner[9]. This study functionally links promoter CGI hypermethylation with oncogenic transformation, demonstrating a causal relationship. Nevertheless, how de novo DNA methyltransferase activity is preferentially targeted to specific regions of the genome in the context of aberrant cancer methylation remains largely a mystery.

It has been hypothesised that cancer cells follow an evolutionary trajectory towards a stem cell state, which allows both self-renewal and differentiation[10], and more recently, cancer-related mutations have been identified in naïve human embryonic stem cells (hESCs)[11].

Here we identify cancer-like DNA methylation changes during primed to naïve hESC resetting using the recently developed NANOG/KLF2 overexpression + 2iLGö method (comprising two small-molecule inhibitors of MEK and GSK3β, human recombinant leukaemia inhibitory factor, and a pan-PKC inhibitor)[12]. Our system provides a unique opportunity to investigate the mechanism of DNA hypermethylation in human cells in a temporal manner, and sheds light on the role of the transcription factor and pluripotency networks in driving cancer-like DNA hypermethylation.

## Results

**Naïve resetting induces CGI promoter hypermethylation**. To investigate the kinetics of the changing DNA methylation landscape between primed and naïve hESCs, we transitioned primed hESCs to the naïve state as previously described[12], by inducing NANOG/KLF2 transgenes with doxycycline. We also captured the two intermediary states, termed 'early transition' and 'late transition' when the cells are in 2iL+dox or 2iL+Gö, respectively (Fig. 1a). We see global DNA demethylation of the genome in naïve cells as reported previously[12], measured by the Infinium MethylationEPIC array and mass spectrometry (Supplementary Figs 1a–c). The loss of 5-methylcytosine (5mC) is gradual and is accompanied by the loss of its oxidation product, 5-hydroxymethylcytosine (5hmC) (Supplementary Fig. 1a). Interestingly, while the majority of the genome is demethylated, we observe hypermethylation of a subset of CpGs (an increase of >10% methylation compared to primed hESCs), exemplified by the HOXA cluster (Fig. 1b, c, Supplementary Fig. 1d). This gain in methylation is evident as cells go through the early transition of resetting, with a peak of hypermethylated CpGs as the cells go through the late transition of resetting (Fig. 1b, c). Although the peak of hypermethylation coincides with the cells being transitioned into 2iL+ Gö conditions, the abundance of hypermethylation is independent of the addition of Gö (Supplementary Fig. 1e), indicating a time-dependent accrual of DNA methylation

instead. As the cells stabilise in the naïve state, we observe maintenance of a proportion of hypermethylated sites, while some CpGs show only a transient gain in methylation (Fig. 1b). The reproducibility of the hypermethylation during the resetting process is apparent from the strong overlap between hypermethylated sites across biologically independent MethylationEPIC arrays (with 2 or 3 cell populations assayed within each array) and when compared to published whole-genome bisulfite sequencing (WGBS) data, suggesting that the site-specific gain in methylation is not random, and likely has a biological function (Supplementary Figs 1f, g). Moreover, as primed hESCs and hESCs during the early transition of resetting proliferate and cycle at comparable rates as measured by loss of bromodeoxyuridine (BrdU), the site-specific gain in methylation upon resetting is the result of an active process rather than the selection of an existing subpopulation of cells (Supplementary Fig 1h).

We next sought to investigate the genomic context with which hypermethylation occurs. We utilised the Encyclopaedia of DNA elements (ENCODE) ChIP-seq datasets for the H1 primed hESC cell line and overlapped them with resetting-associated hypermethylated probes. We observed that hypermethylated probes are enriched within regions marked by H3K4me1/2/3 and H3K27me3 in primed hESCs (Fig. 1d). The majority of these fall within regions marked by bivalent histone modifications, defined by co-occurrence of H3K4me3 and H3K27me3 (Fig. 1e). Bivalency typically marks regulatory regions (promoters and enhancers with overlapping transcription factor binding sites)[13], and as expected, we saw a striking overlap with CGIs and regulatory regions, which is not the case for hypomethylated probes (Fig. 1f, g). In addition, the majority of the hypermethylated probes reside within ChromHMM predicted poised promoters (Fig. 1h). To validate our results, we also re-analysed published whole-genome bisulfite sequencing data for primed and naïve hESCs[12], and identified 26,625 regions (300 bp each) that were hypermethylated (>10% increase in naïve vs primed) in naïve hESCs. In agreement with our Illumina MethylationEPIC array data, we observed enrichment of these regions within loci marked by bivalency in primed hESCs, with a bias towards regulatory regions and CGIs (Supplementary Figs 2a–e). This strongly reinforces the highly reproducible nature of the DNA hypermethylation that occurs upon resetting of primed hESCs to naïve pluripotency.

To eliminate the possibility that the hypermethylation is simply an artefact of the NANOG/KLF2 + 2iLGö in vitro resetting system, we compared 2iLGö naïve hypermethylated regions to hypermethylated regions identified in naïve cells generated using the alternative methods of naïve hESC generation and those in the human inner cell mass (ICM)[14–17]. We saw a significant overlap between the hypermethylated regions in each of these datasets compared to the 2iLGö naïve cells, in both cases also enriched within H3K4me1/2/3 and H3K27me3 regions in primed hESCs (Supplementary Fig 3a–g). Interestingly, we do not detect any hypermethylation in mouse embryonic stem cells cultured in 2i compared to those cultured in serum[18], and the hypermethylation present in the mouse ICM[19] is far less extensive and does not enrich at bivalent regions (Supplementary Fig 3h–j). Overall from these results, we can conclude that the hypermethylation pattern is a feature of in vitro naïve human pluripotency and recapitulates the in vivo relationship between the ICM and the post-implantation embryo.

**Developmental genes are hypermethylated and repressed**. To identify the genes targeted by hypermethylation, we performed gene ontology (GO) analysis of the genes and classified a gene as hypermethylated if it possessed a hypermethylated probe/ region

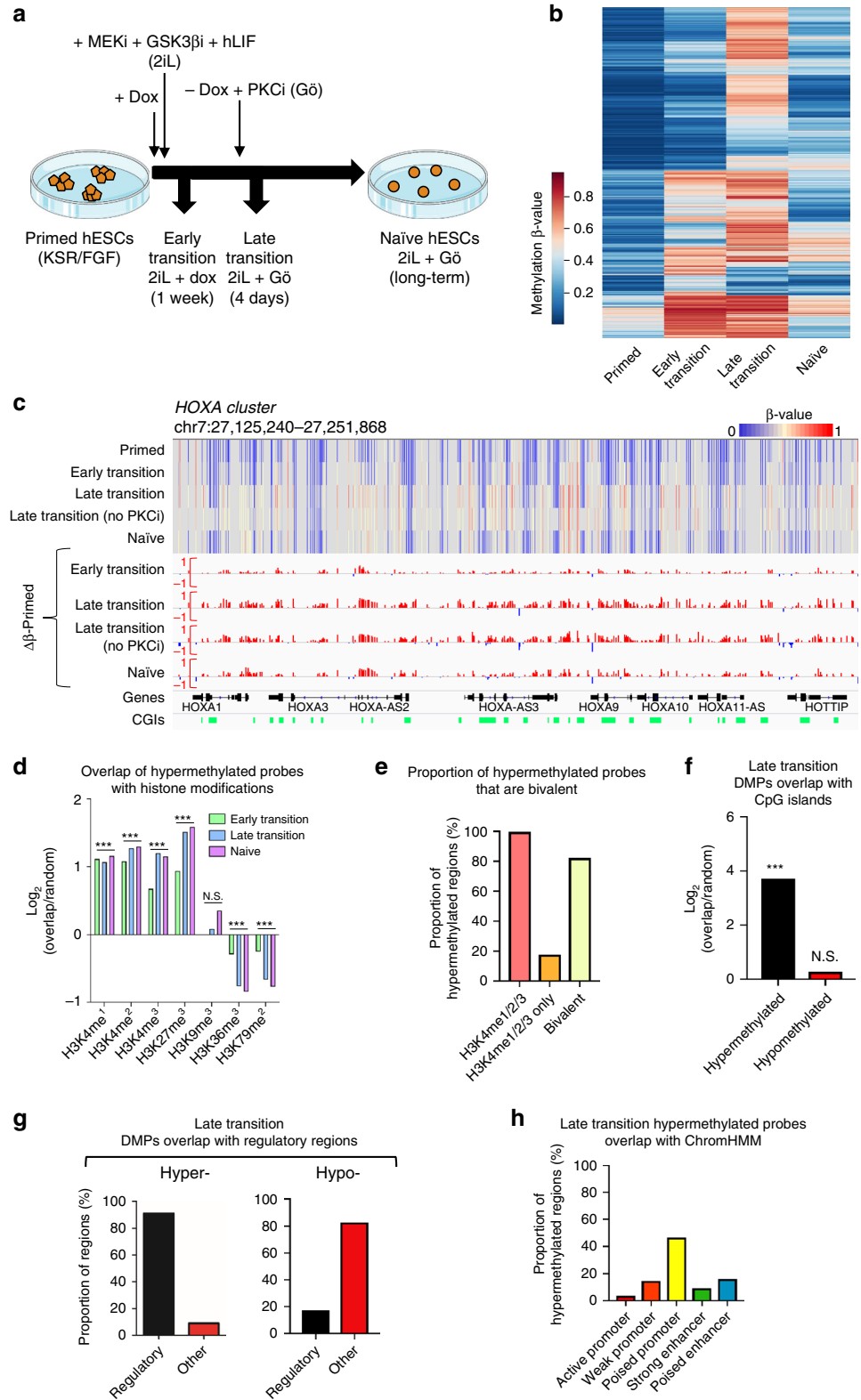

within 1500 bp upstream of its transcription start site (TSS). GO analysis revealed an extensive enrichment of hypermethylated genes in developmental pathways, particularly neuronal development, while hypomethylated genes show much weaker enrichment in pathways involved in cell cycle and metabolism (Fig. 2a, Supplementary Fig. 4a). To investigate whether hypermethylation is associated with gene silencing, we performed

temporal transcriptome analysis of cells during the transition from primed to naïve pluripotency and observed thousands of genes differentially expressed at each stage of the transition compared to primed hESCs, with an enrichment of down-regulated genes in developmental pathways (Supplementary Fig 4b, c). We observed that the average expression of genes which undergo hypermethylation is attenuated in naïve compared

**Fig. 1 Primed to naïve resetting induces bivalent CGI promoter hypermethylation. a** Schematic detailing the model system and time points used in the study. 2iL+dox: 2 small-molecule inhibitors of MEK1/2 and GSK3β (2i), human recombinant leukaemia inhibitory factor (hLIF; collectively 2iL) and doxycycline. 2iL+Gö: 2iL and a pan-protein kinase C inhibitor (PKCi), Gö. hESCs, human embryonic stem cells. **b** Heatmap showing methylation levels of the top 10,000 CpG probes that are differentially methylated ($\Delta\beta > 0.1$, adjPval < 0.05) in the early transition, late transition and naïve hESCs compared to primed hESCs. Methylation β-value is indicated by the colour key. adjPval based on Benjamini–Hochberg adjustment. **c** Genome browser tracks for Infinium MethylationEPIC data capturing a representative hypermethylated locus. The heatmap shows the raw methylation β-values per CpG for each sample, while the subsequent rows show the per-probe difference in methylation for each time point of resetting compared to primed hESCs. CGIs are highlighted in green. **d** Overlap of hypermethylated probes ($n = 91,119$) with regions of histone modification enrichment (obtained from the ENCODE ChIP-seq data for hESC cell line H1: H3K4me1 $n = 139971$; H3K4me2 $n = 73086$; H3K4me3 $n = 33,270$; H3K9me3 $n = 86,122$; H3K27me3 $n = 25,909$; H3K36me3 $n = 35,877$; H3K79me2 $n = 33205$). **e** The proportion of hypermethylated probes ($n = 46,844$ early transition, $n = 91,119$ late transition, $n = 20,297$ naïve) that are marked by H3K4me1/2/3, H3K4me1/2/3 alone, or bivalency (H3K4me3 and H3K27me3). **f** Overlap of late transition hypermethylated probes ($n = 91,119$) and hypomethylated probes ($n = 392,875$) with CpG islands ($n = 30,344$). **g** Overlap of hypermethylated and hypomethylated probes (as in 1f) with ENCODE regulatory regions (promoters or enhancers). **h** Proportion of late transition hypermethylated probes ($n = 91,119$) that overlap with ENCODE predicted promoters and enhancers (as defined by ChromHMM in the hESC cell line H1). DMP, differentially methylated probes. For overlap analysis, data are presented as the $\log_2$ corrected fold increase in the observed overlap compared to the mean overlap of 1000 randomly generated loci. ***$P < 0.001$.

to primed hESCs, but that hypermethylated genes are characterised by low average expression in primed hESCs (Fig. 2b, c). Hypermethylation may play a functional role in these cells by contributing to downregulation of developmental pathways, perhaps enhancing pluripotency and blocking differentiation by providing a more stable gene repression mechanism[13]. Intriguingly, we observe a subset of genes, notably the HOX gene family, that become hypermethylated and are upregulated upon hESC resetting (Fig. 2d), despite no evidence of 5mC oxidation to 5hmC at their promoters (Fig. 2e). Interestingly, the methylation dynamics of these HOX genes do not differ from genes that are hypermethylated and downregulated (Fig. 2c, d, Supplementary Fig. 4d). This suggests that there may be population heterogeneity during resetting, whereby a subpopulation of cells which do not undergo HOX promoter hypermethylation exhibit upregulation of the genes.

**DNMT3A controls early de novo methylation.** We next sought to identify the epigenetic regulators that are responsible for the deposition of de novo DNA methylation. Of the de novo DNMT3 family of DNA methyltransferases, *DNMT3B* is highly expressed but is transiently downregulated upon resetting. The mRNA level of *DNMT3A* is transiently upregulated (Supplementary Fig. 5a), though this is not reflected in the protein level (Supplementary Fig. 5g). The catalytically inactive *DNMT3L* is upregulated (Supplementary Fig. 5a) and considered a marker of naïve pluripotency[20]. We generated constitutive knockdown primed hESC cell lines using two short hairpin RNAs (shRNAs) targeting each of the three genes, as well as one specifically targeting the long isoform of *DNMT3A*, known as *DNMT3A1* (Supplementary Fig. 5b). We subjected each of the cell lines to resetting until the early transition, at which stage hypermethylation is already detectable, and thereafter to the late transition. In the early transition, knockdown of *DNMT3B* and *DNMT3L* had little impact on the level of methylation (Fig. 3a, Supplementary Fig. 5c). Knockdown of *DNMT3A*, however, was able to abolish hypermethylation (Fig. 3a). A recent study demonstrated isoform-specific recruitment of DNMT3A1 to bivalent CGIs in mouse embryonic stem cells[21]. However, specific knockdown of *DNMT3A1* had no impact on the level of hypermethylation (Supplementary Fig. 5d), suggesting that the more dominantly expressed *DNMT3A2* carries out de novo methylation early during resetting, independently of DNMT3L. It is worth noting, however, that during the late transition of resetting, both DNMT3A and DNMT3B knockdown cells show a partial reduction in methylation compared to the control cells (Fig. 3b). This indicates that DNMT3B, which by this stage is transcriptionally expressed at a higher level than during the early

transition (Supplementary Fig. 5a), contributes to hypermethylation along with DNMT3A. Additionally, when we reset either DNMT3A or DNMT3B knock down cells to the naïve state, we see a reduction in the pluripotency of the cells as measured by their alkaline phosphatase activity as well as reduced expression levels of naïve pluripotency genes in DNMT3A but not DNMT3B knock down cells (Fig. 3c, d, Supplementary Fig. 5e). As both DNMT3A and DNMT3B seem to contribute to hypermethylation during the late transition of resetting, we additionally generated a double knock down cell line of the two genes in primed hESCs and subjected the cells to resetting to the naïve state (Supplementary Fig 5f). The DNMT3A/DNMT3B double knock down cells show significantly reduced levels of hypermethylation during the early transition as well as in naïve hESCs, and they exhibit reduced pluripotency as measured by their alkaline phosphatase activity and expression levels of naïve pluripotency genes, though TFCP2L1 and KLF4 do not change significantly (Fig. 3e–g). Collectively, this indicates a putative role of the de novo methyltransferases in stabilisation of the naïve pluripotent state.

Aside from the de novo methyltransferases, we also hypothesised that loss of ten-eleven translocation (TET) enzymes may be the cause of bivalent CGI hypermethylation[22,23]. TET1 is expressed in primed hESCs and subsequently downregulated at the protein level as hESCs progress through the early transition of resetting (Supplementary Fig 6a, b). We generated TET1-overexpressing primed hESCs (Supplementary Fig 6c, d) and subjected them to resetting until the early transition. We measured DNA methylation at selected target loci but observed no change in the levels of DNA methylation, though TET1 is modestly overexpressed (Supplementary Fig. 6e). These data indicate that hypermethylation of bivalent loci upon resetting is independent of TET1 loss.

To determine whether de novo methylation is strictly correlated with the acquisition of naïve pluripotency during resetting, we used a published cell-surface marker, SUSD2, that has been shown to separate naïve from primed hESCs during resetting[24], and is able to identify increasing numbers of naïve hESCs during the transition to the naïve state (Supplementary Fig. 7a). We sorted SUSD2+ and SUSD2- cells during the early and late transition of resetting and used a targeted approach to measure DNA methylation at selected resetting-associated hypermethylated loci (Fig. 4a). We observed no significant difference in the level of hypermethylation in SUSD2+ and SUSD2- fractions, though SUSD2+ cells displayed higher expression of naïve pluripotency genes (Fig. 4b, c, Supplementary Fig. 7b). In conjunction with the observation that reduced expression of de novo methyltransferases during resetting impacts naïve pluripotency, this indicates that while de novo methylation may be important for the stability of the naïve

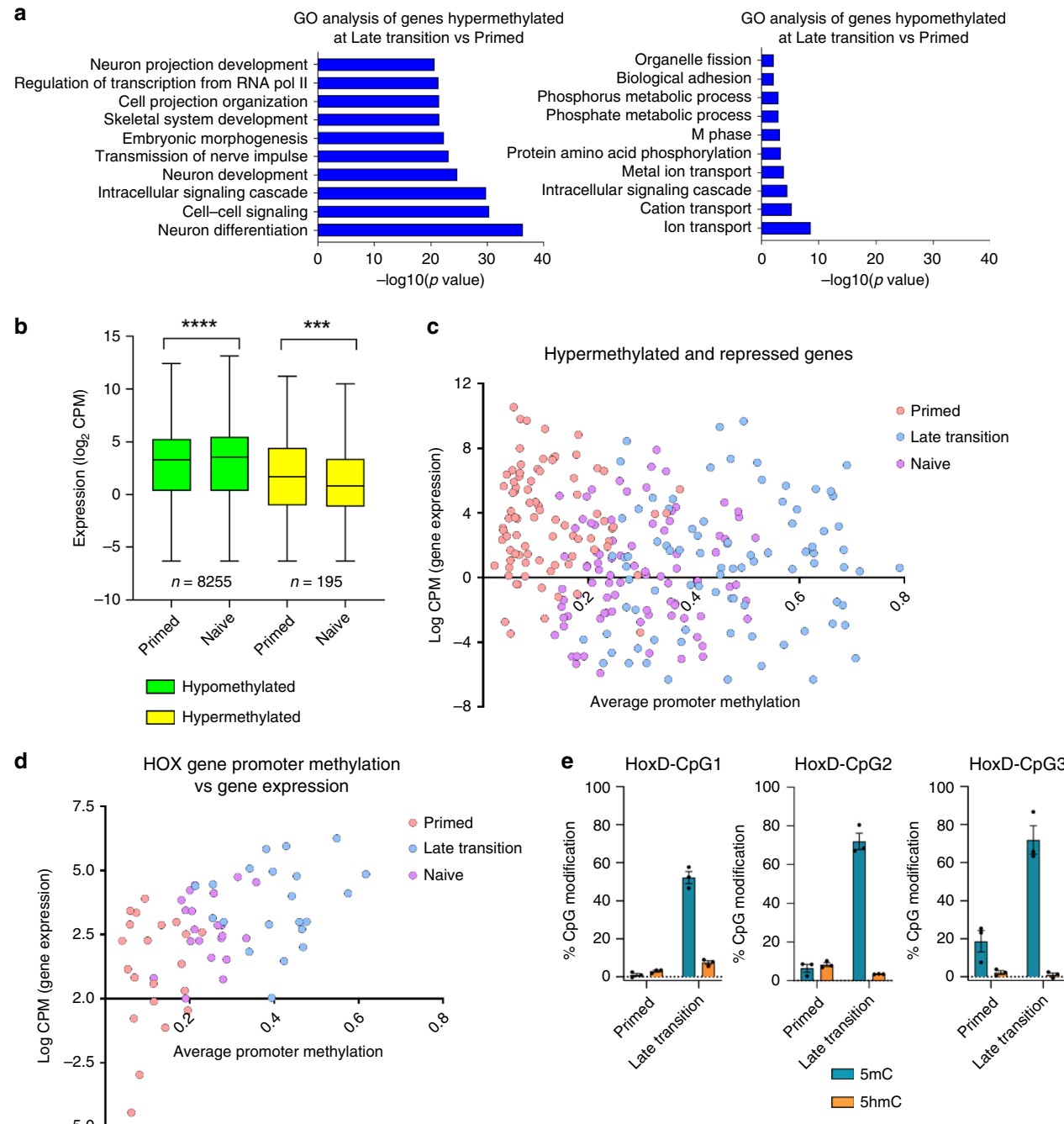

**Fig. 2 Resetting results in hypermethylation and repression of developmental genes. a** GO term analysis of hypermethylated and hypomethylated genes at the late transition of resetting compared to primed hESCs. A gene was classified as hypermethylated if a hypermethylated probe was present within 1500 bp upstream of the TSS. **b** Average gene expression for genes that are hypermethylated (average promoter methylation $\Delta\beta > 0.1$) or hypomethylated (average promoter methylation $\Delta\beta < 0.1$) in naïve compared to primed hESCs. Boxes represent the median and interquartile range and error bars represent the maximum and minimum values. Statistical significance determined via two-tailed paired Wilcoxon test. ***$P < 0.001$, ****$P < 0.0001$. CPM, Counts per Million. **c** Scatter plot of genes that are hypermethylated and downregulated showing the average promoter methylation (average $\beta$-value of CpG probes within 1500 bp of TSS) versus the log2 CPM (counts per million) for each gene from RNA-seq data. Data for each individual time point are indicated by the colour key. **d** Scatter plot showing the average promoter methylation of 21 HOX genes (average $\beta$-value of CpG probes within 1500 bp of TSS) versus the log2 CPM for each gene from RNA-seq data. Data for each individual time point are indicated by the colour key. **e** GlucMS-qPCR in primed and late transition hESCs showing the percentage of 5mC (blue) and 5hmC (orange) per CpG. Bars represent the mean of three biological replicates and error bars represent the SEM. Source data are provided as a Source Data file.

pluripotent state, it is insufficient for the acquisition of naïve pluripotency. Collectively, this suggests that the hypermethylation is more a feature of the heterogeneous intermediate population of partially reset cells.

**Bivalent CGIs that lose H3K4me3 gain DNA methylation**. To investigate the mechanism of bivalent CGI hypermethylation, we first classified bivalent regions as those possessing H3K4me3 and H3K27me3 peaks in primed hESCs utilising the ENCODE ChIP-

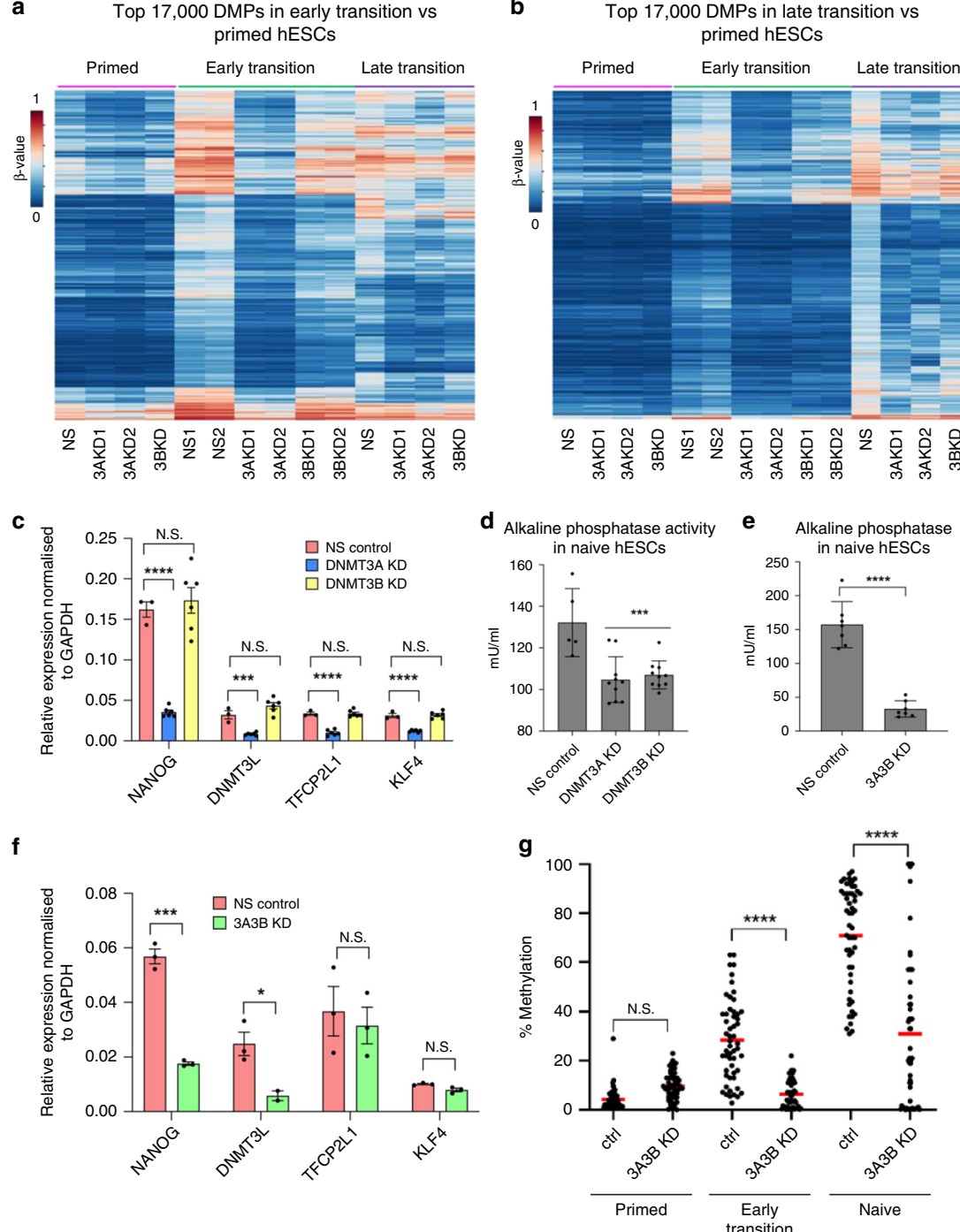

seq dataset. We observed that only 41% of these loci gain DNA methylation upon resetting (Fig. 5a), indicating that the presence of bivalent chromatin is not sufficient for acquiring de novo DNA methylation upon resetting. We therefore divided all bivalent regions in primed hESCs into those that do and do not gain methylation during resetting (hypermethylated regions defined as >10% increase in naïve vs primed hESC). Taking the nearest gene to each region (within 1500 bp of the TSS), GO analysis of the bivalent hypermethylated group showed a strong enrichment for developmental pathways, while the bivalent non-hypermethylated group showed much lower enrichment of other biological processes (Fig. 5a). This points towards common regulation of the developmental genes that exhibit hypermethylation during resetting.

We hypothesised that there are intrinsic differences between bivalent regions that do and do not gain methylation, which both begin with similar chromatin states. As DNA methylation and H3K4me3 are known to be mutually exclusive[25], we performed ChIP-qPCR of H3K4me3 at bivalent DNA regions, across the time course of resetting. We observed a loss of H3K4me3 at bivalent regions that become hypermethylated, whilst bivalent non-hypermethylated regions retain their levels of H3K4me3 (Fig. 5b). In contrast, the levels of H3K27me3 exhibit little change, despite the presence of DNA methylation (Fig. 5c). While H3K27me3 and DNA methylation are considered to be mutually exclusive at CpG rich regions during development[26], co-existence of the two modifications has previously been reported[27]. It is

**Fig. 3 Early de novo methylation is dependent on DNMT3A. a**, **b** Heatmaps showing methylation levels in control and DNMT3A/B knock down samples during resetting. Heatmap shows the top 17,000 CpG differentially methylated probes (DMP; Δβ > 0.1, p < 0.05) in the early transition (Fig. 3a) and late transition (Fig. 3b) compared to primed hESCs (in wild type early transition (Fig. 3a) or late transition (Fig. 3b) compared to primed hESCs identified in analysis shown in Fig. 1b). Methylation β-value is indicated by the colour key. **c** qRT-PCR for naïve pluripotency genes in control and knockdown cells, in naïve hESCs. Bars represent mean of three biological replicates and error bars represent the SEM. Statistical difference between samples was calculated by a one-way ANOVA with a Bonferroni post hoc test compared to the control. Human GAPDH was used to normalise expression. **d** Alkaline phosphatase activity in knock down and control naïve hESCs. Data shown are the mean of two biological replicates (independent shRNA KD), each with five technical replicates. Error bars represent SEM. Statistical difference between samples was analysed by a one-way ANOVA with a Bonferroni post hoc test compared to the control. **e** Alkaline phosphatase activity in DNMT3A/DNMT3B double knock down and control naïve hESCs. Data shown are the mean of five replicates. Error bars represent SEM. Statistical difference between samples was analysed by a two-tailed Student's unpaired t test. **f** qRT-PCR for naïve pluripotency genes in control and DNMT3A/DNMT3B double knockdown naïve hESCs. Bars represent the mean of three biological replicates and error bars represent the SEM. Statistical difference between samples was calculated by a two-tailed Student's t test. Human GAPDH was used to normalise expression. **g** Plot showing % methylation in DNMT3A/DNMT3B double knock down and control cells during resetting. Each dot represents % methylation of single CpGs (n = 57) from 4 genomic regions analysed by targeted bisulfite sequencing. Red bars represent mean methylation for each sample. Statistical difference between samples was analysed by a Kruskal–Wallis test, with Dunn's multiple comparisons post hoc test. For all panels, *P < 0.05, **P < 0.01, ***P < 0.001, ***(P < 0.0001), N.S. denotes not significant (p > 0.05). Source data are provided as a Source Data file.

known that loss of H3K4me3 is permissive to the gain of DNA methylation, but this on its own cannot explain the specific gain of methylation at these regions, as both DNMT3A and DNMT3B possess an ADD domain capable of mediating the interaction of the enzymes with unmethylated H3K4[25,28–30], and several loci that have been shown to lose H3K4me3 in naïve hESCs do not undergo hypermethylation[14]. Additionally, despite comparable absolute protein levels of DNMT3A and DNMT3B, as measured by mass spectrometry (Supplementary Fig. 5g), only DNMT3A deposits DNA methylation during the early transition of hESC resetting. Moreover, the strong bias towards developmental genes suggests that the gradual hypermethylation is not a stochastic process.

**Transcription factors influence hypermethylation.** Our data led us to hypothesise that an additional player, likely a DNA-binding factor, facilitates DNMT3A-mediated hypermethylation in the early stages of hESC resetting. To understand the unique properties of the bivalent hypermethylated group, we performed differential motif analysis of these regions, with the bivalent non-hypermethylated regions as a control set. We identified a number of motifs corresponding to DNA-binding transcription factors enriched at regions that undergo hypermethylation (Fig. 6a). To test whether these proteins are expressed, we performed total proteomics of primed and early transition hESCs and identified proteins that were upregulated during the early transition compared to primed hESCs (Fig. 6b). Through cross-comparison of the two analyses, we short-listed two candidate transcription factors, SOX15 and NFKB1, which are upregulated during early resetting and show an enrichment of binding sites at hypermethylated regions (Fig. 6a, b). We identified an additional two candidate transcription factors, FOXC1 and ZFHX3 (Fig. 6a), which were transcriptionally upregulated during the early transition based on RNA-seq data but not detected in any samples by proteomic analysis, likely due to technical limitations of the method in detecting nuclear transcription factors[31]. We generated constitutive knockdown cell lines using two shRNAs targeting each of the four candidate genes (Supplementary Fig. 8a) and subjected each of the cell lines to resetting until the early transition. We measured the expression of naïve pluripotency genes to test whether the knock down cells undergo resetting similarly to control cells, and found that their expression is not significantly altered in the knockdowns compared to control cells (Supplementary Fig. 8b). Strikingly, upon resetting, knock down of each of the transcription factors reduced the level of hypermethylation at target loci analysed, suggesting that the transcription factor network active during the early transition of

resetting is involved in bivalent promoter CGI hypermethylation (Fig. 6c). The impact of each of the knockdowns on DNA methylation is higher in regions where at least one of the highly expressed transcription factors are predicted to bind compared to sites that are not bound by any of the four transcription factors (Fig. 6c). Interestingly, the reduction in methylation is observed in each of the transcription factor knockdowns at SOX15 and NFKB1 predicted binding sites, indicating that the impact is not limited to the specific binding sites for each transcription factor (Fig. 6c). This points to a network synergy in preferentially mediating de novo methylation at these sites. As the reduction in DNA hypermethylation upon transcription factor knockdown is only partial; however, this suggests that additional mechanisms are also at play.

To test whether signalling changes associated with factors required for induction of the naïve state induction could explain such a mechanism, we conducted hESC resetting until the early transition, each time removing one of these factors. Resetting in the absence of the MEK inhibitor or GSK3β inhibitor or concomitant removal of both inhibitors still resulted in hypermethylation at target loci analysed (Fig. 6d), suggesting that hypermethylation may be coordinated by the overexpressed NANOG and KLF2 or the associated pluripotency network. Collectively, these data indicate that upon reprogramming hESCs to the naïve state, hypermethylation is driven by the transcription factor network that becomes active upon resetting, and that this is synchronised by the core pluripotency network.

**Resetting-associated hypermethylation is mirrored in cancer.** De novo DNA methylation of bivalent chromatin in the context of a hypomethylated genome has also been reported in cancer cell lines and primary tumours[32,33]. To investigate the link between the hypermethylation patterns associated with hESC resetting and the re-emergence of such patterns in cancer, we compared hypermethylated CpGs at each stage of resetting with regions previously identified as hypermethylated in B-cell chronic lymphocytic leukaemia (B-CLL)[34]. We observed that the most substantial overlap is found between hypermethylated CpGs associated with the late transition of resetting and B-CLL hypermethylation, corresponding to the peak of hypermethylation we observe during reprogramming (Supplementary Fig. 9a). Interestingly, we also see a more significant overlap between hypermethylated regions in B-CLL or colon cancer[34,35] and resetting-associated hypermethylated CpGs with a low basal methylation level (<5%) in primed hESCs (Supplementary Figs 9b, c). These data demonstrate a substantial overlap between naïve stem cell and cancer hypermethylation.

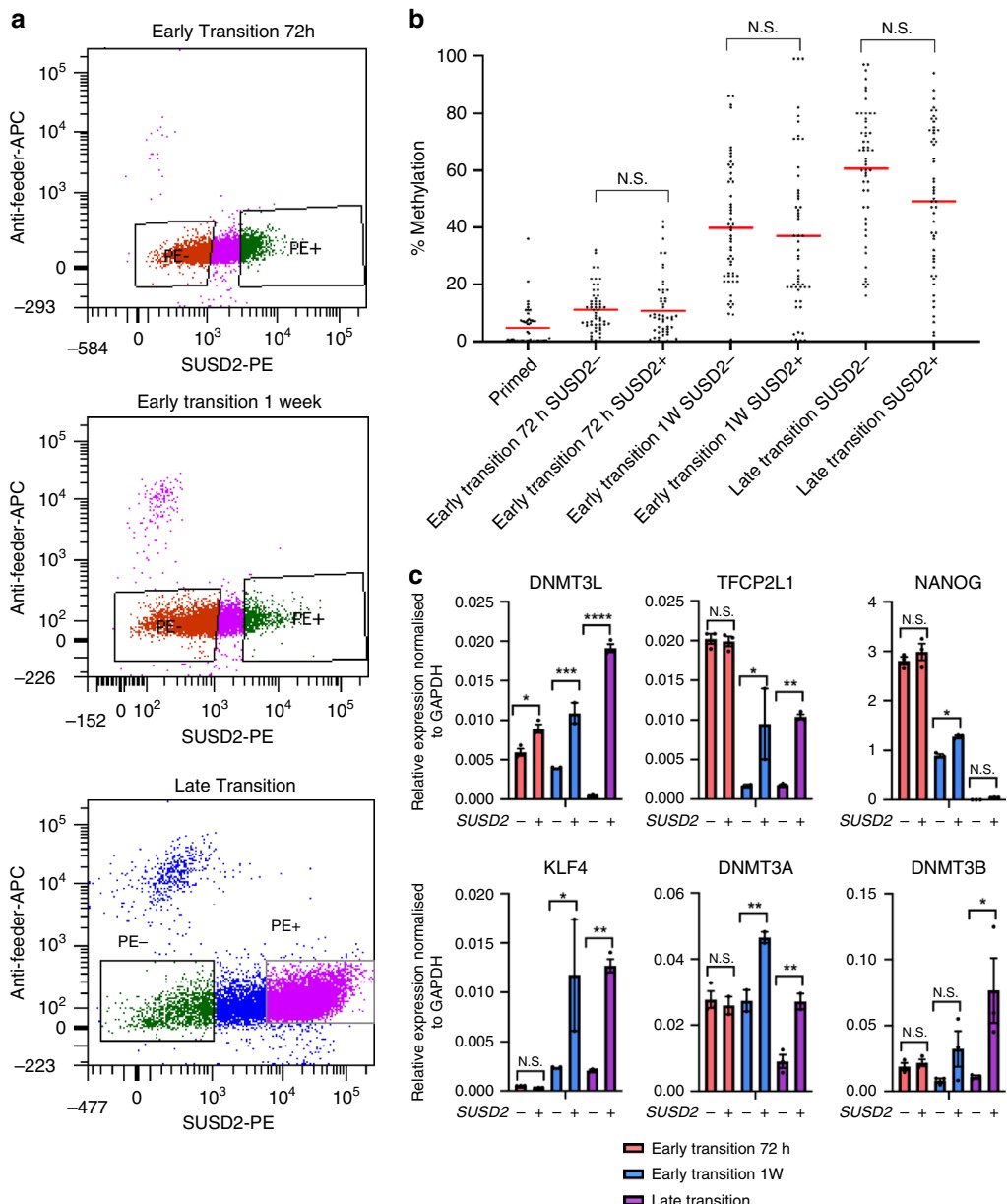

**Fig. 4 Hypermethylation is a feature of resetting and is not restricted to emerging naïve cells. a** Flow cytometry dot plots showing SUSD2-PE staining on the x-axis against anti-feeder-APC staining on the y-axis for hESCs during the early and late transition. Boxes indicate SUSD2+ and SUSD2- cell populations that were sorted. **b** Plot showing the % methylation in primed, early transition and late transition hESCs, for SUSD2+ and SUSD2− cell populations. Each dot represents the methylation % of single CpGs ($n = 57$) from four genomic regions analysed by targeted bisulfite sequencing, and the red bars represent the mean methylation level for each sample. Statistical difference between samples was analysed by a Kruskal–Wallis test, with Dunn's multiple comparisons post hoc test. N.S. denotes not significant ($p > 0.05$). **c** qRT-PCR for naïve pluripotency genes in SUSD2+ and SUSD2− cell populations early and late transition hESCs. Bars represent the mean of three technical replicates and error bars represent SEM. Statistical significance between samples was analysed with a one-way ANOVA with a Bonferroni post hoc test comparing all samples to each other. Human GAPDH was used to normalise expression. *$P < 0.05$, **$P < 0.01$, ***$P < 0.001$, ****$P < 0.0001$, N.S. denotes not significant. Source data are provided as a Source Data file.

We further hypothesised that cancer cells may recapitulate the dichotomy between bivalent genes that do and do not become hypermethylated. To test this, we compared resetting-associated bivalent hypermethylated and non-hypermethylated CpGs with data from the cancer genome atlas (TCGA) pan-cancer atlas[36,37]. We found a significant gain in methylation between normal and cancer tissue for bivalent CpGs identified as hypermethylated during the resetting process, compared to unmethylated bivalent CpGs (Fig. 7a, Supplementary Fig. 10a). This was consistent across all cancer types analysed. In addition, when we defined CpGs that were hypermethylated compared to normal tissues across cancer

types, we found that they were significantly enriched in hESC hypermethylated bivalent CpGs (Fisher's test $p$-value $< 2 \times 10^{-16}$, percentage fold-change 11.13, Fig. 7b). Similar results were also observed when defining hypermethylated CpGs for each of the nine individual tumour types (Fig. 7b). Together these results indicate that resetting-associated hypermethylation parallels pan-cancer hypermethylation, though the individual CpGs methylated in each cancer type vary[38]. We see no further enrichment of resetting-associated hypermethylated CpGs with more advanced stages of cancer (Supplementary Fig. 9d), or with datasets derived from metastatic tissues (Supplementary Fig. 9e).

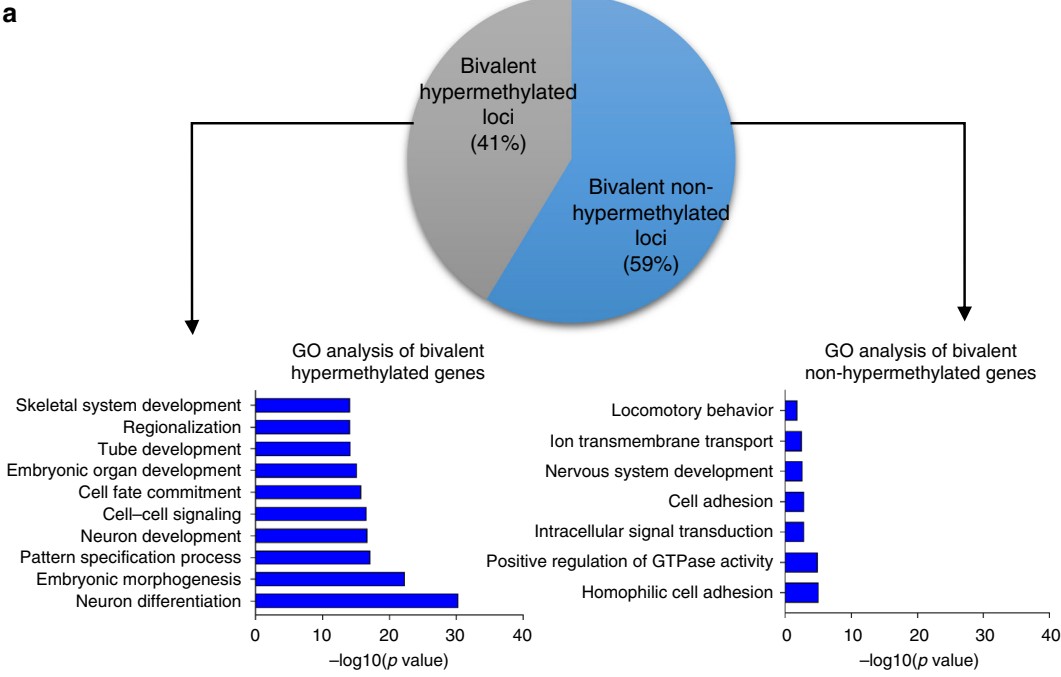

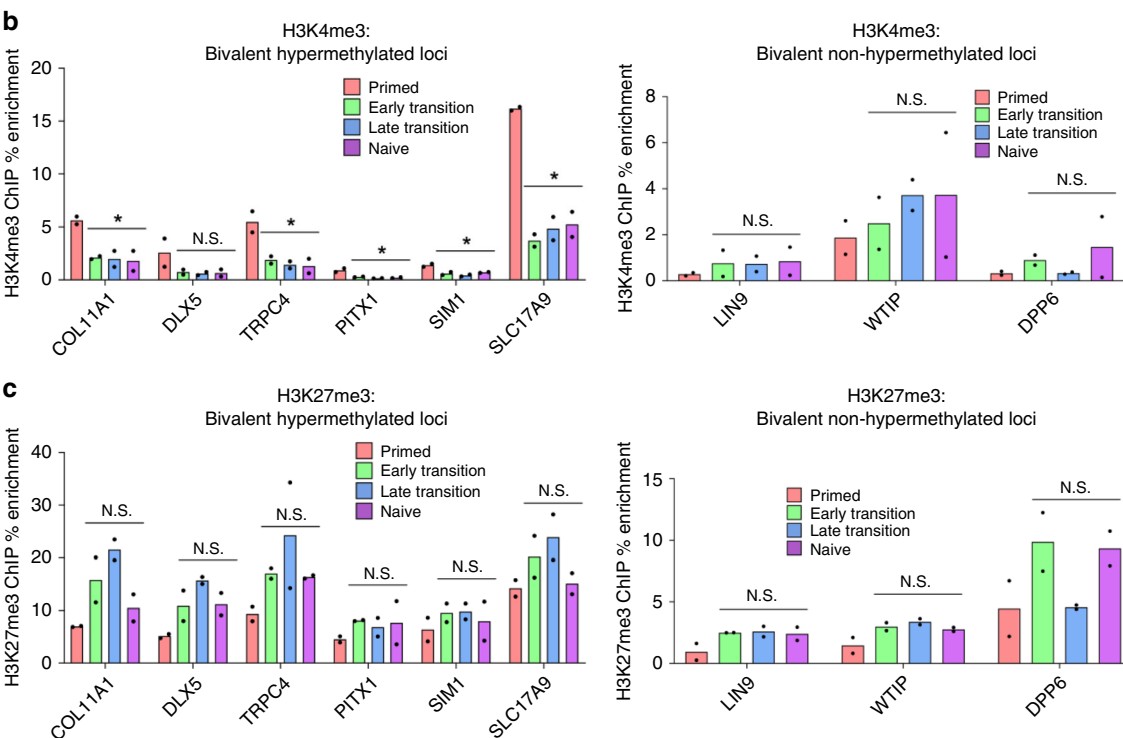

**Fig. 5 Bivalent CGIs that lose H3K4me3 gain DNA methylation. a** Pie chart showing the proportion of bivalent sites that do and do not gain methylation in naïve hESCs, and gene ontology analysis of these bivalent hypermethylated and non-hypermethylated genes respectively. **b** ChIP-qPCR enrichment of H3K4me3 and (**c**) H3K27me3 are shown for six candidate bivalent regions (possessing both H3K4me3 and H3K27me3 histone modifications in primed cells) that become hypermethylated, and for three candidate bivalent regions that fail to become hypermethylated during resetting (bottom row). Data are shown as the signal enrichment relative to the input sample with bars representing the mean of two independent experiments. Statistical difference between samples was analysed by a one-way ANOVA test, with Bonferroni post hoc test of each time point compared to primed hESC. * indicates $p < 0.05$. Source data are provided as a Source Data file.

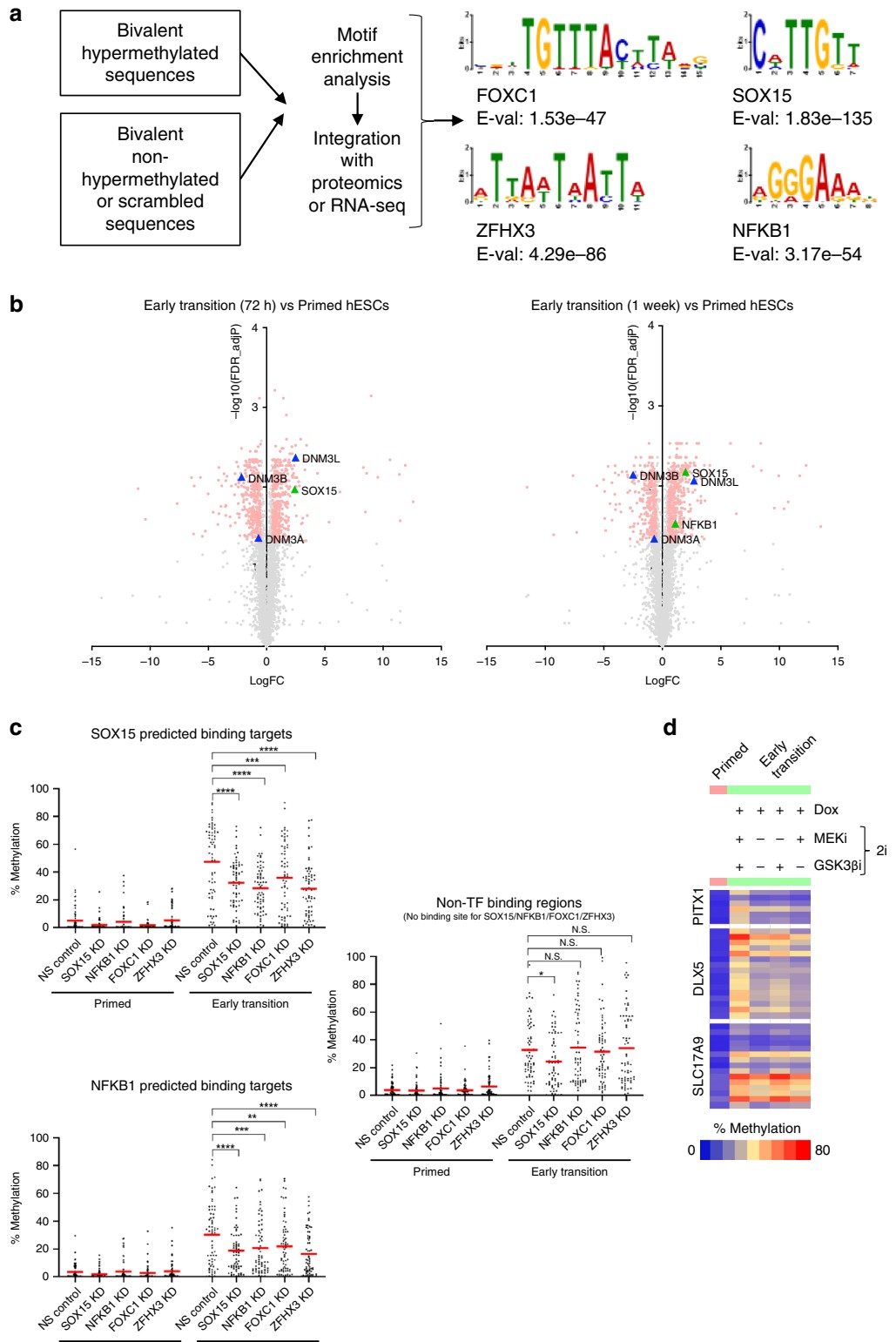

To further investigate the parallels between hypermethylation in naïve hESCs and cancer, we asked whether the regions hypermethylated in naïve hESCs were also marked by H3K27me3 in normal human tissues that give rise to cancers. By examining H3K27me3 ChIP-seq from eight normal tissues, we found that the regions surrounding hypermethylated CpGs had significantly higher levels of H3K27me3 than non-hypermethylated CpGs from naïve hESCs (Fig. 7c, Supplementary Fig. 10b). This included the HOX loci which we had seen to be strongly hypermethylated in naïve hESCs as well as loci which are previously reported as gaining DNA methylation in cancer such as SFRP1 (Fig. 7d)[8].

Together, these results demonstrate that resetting of primed hESCs to naïve pluripotency and cancer follow similar epigenetic trajectories, both involving the acquisition of DNA methylation at developmental gene promoters marked by H3K27me3. There is

**Fig. 6 Transcription and pluripotency factors influence hypermethylation. a** A selection of the transcription factors with motifs enriched in bivalent hypermethylated regions, with bivalent non-hypermethylated regions used as a background control. Motif analysis was performed using the analysis of motif enrichment (AME) tool on the MEME suite. **b** Volcano plots showing the difference in protein expression in early transition (72 h and 1 W) hESCs compared to primed hESCs (Supplementary Table 6). Each dot represents the log2 fold change based on three biological replicates. Statistical difference between samples was analysed by a two-tailed student's $t$ test, corrected for multiple testing. Red dots indicate statistically significant changes (adjP < 0.05). Proteins of interest are highlighted with coloured and labelled symbols. **c** Plot showing the % methylation for four different transcription factor knockdowns and a non-silencing control. Data for each sample are an average of two independent shRNA knockdowns. Each dot represents the methylation % of a single CpG analysed by targeted bisulfite sequencing, and the red lines represent the mean % methylation level for each sample from five genomic regions (SOX15 targets; $n = 64$ CpGs), or four genomic regions (NFKB1 regions; $n = 67$ CpGs, or regions without TF binding sites for any of the 4 TFs; $n = 66$ CpGs). Statistical difference between samples was analysed by a two-way ANOVA test, with Bonferroni post hoc test of each TF knock down compared to the control. *$P < 0.05$, **$P < 0.01$, and ***$P < 0.001$, ****$P < 0.0001$. N.S. denotes not significant. **d** Targeted bisulfite-sequencing of three genomic regions. Each square represents the methylation % indicated by the colour key of a single CpG. The first column represents data from primed hESCs, and the subsequent columns represent data from early transition hESCs cultured in a variety of culture conditions indicated by the ± symbols above. Source data are provided as a Source Data file.

growing evidence in the literature regarding the acquisition of stem-like properties and expression of pluripotency genes in cancers[10,39]. This makes it intriguing to speculate that the pluripotency network plays a key role in establishing cancer hypermethylation. Considering the causal relationship between hypermethylation and acceleration to one-step transformation[9], this warrants further investigation of means to disrupt de novo accumulation of DNA methylation at CGI promoters.

**Discussion**

We propose the concept that reprogramming events during primed to naïve resetting are a fundamental feature of human cancers and possibly a very early step in cancer evolution. Although naïve stem cells and cancer cells seem developmentally distant, our data support the hypothesis that cancers follow an evolutionary trajectory towards an embryonic stem cell state, which allows both self-renewal and differentiation[10].

We demonstrate the dynamic acquisition of DNA methylation, primarily at CGI promoters, upon the transition from primed to naïve pluripotency in hESCs. We observed the highest levels of hypermethylation in the heterogeneous population of cells present during the late transition of resetting. It is worth noting that during the late transition of resetting, there is population heterogeneity as demonstrated by the study of HOX gene promoters, as well as heterogeneous expression of naïve stem cell markers as exemplified by the SUSD2 expression, but the hypermethylation and expression of naïve stem cell markers do not strictly co-occur. It seems that hypermethylation is a feature of this heterogeneous intermediate population of partially reset cells, and it is partially maintained as the cells stabilise in the naïve state.

We demonstrate that the hypermethylation that we observed upon the transition from primed to naïve pluripotency mirrors the frequently observed aberrant hypermethylation in human cancers. Such parallels have been drawn previously in other mammalian species and developmental contexts[40]. However, the data we present here demonstrate hypermethylation conserved across in vitro and in vivo human pluripotency[17], strengthened by its reproducibility across multiple in vitro resetting methods[14,16]. Moreover, it is notable that we do not observe comparable hypermethylation in the mouse ICM or in in vitro mouse ESCs cultured with 2i inhibitors. This observation has potential implications for making inferences with regards to epigenetic processes between species, both in development and in the study of cancer, as has been noted previously[41].

We have observed that bivalent loci are almost exclusively susceptible to DNA hypermethylation during the transition to naïve pluripotency. However, not all bivalent loci become hypermethylated, suggesting that the presence of bivalent chromatin is not the only factor required for acquiring de novo DNA methylation upon resetting. Several studies have demonstrated

that the aberrant gains of DNA methylation observed in cancer also occur at H3K27me3-marked loci[42–44]. Loci that gain DNA methylation in cancer are also enriched in sets of loci that are bivalent in embryonic stem cells[33,42,45]. The mechanistic basis of this relationship is unclear. Although DNMTs can interact with EZH2 which is responsible for deposition of H3K27me3[46], recruitment of DNMT3A by PRC2 is not sufficient to trigger de novo DNA methylation[47]. It is particularly noteworthy that bivalent loci that undergo hypermethylation both upon resetting and in the context of cancer belong to developmental pathways[33], distinguishing them from other bivalent loci that do not gain methylation despite having a comparable starting chromatin configuration.

Our data indicate that DNMT3A is responsible for DNA hypermethylation during the early transition of primed to naïve resetting, and that both DNMT3A and DNMT3B contribute to hypermethylation during the late transition of resetting. At this stage, the reduction in methylation is only partial upon knockdown of either enzyme, therefore alternative factors such as the putative de novo activity of DNMT1 cannot be excluded[48]. A knockdown of either DNMT3A or DNMT3B impacted the stability of the naïve state as measured by alkaline phosphatase activity, with a knockdown of DNMT3A additionally disrupting the naïve transcriptional network. Interestingly, a double knockdown of DNMT3A and DNMT3B had an even greater impact on the naïve state as measured by alkaline phosphatase activity, suggesting a synergy between the two de novo methyltransferases in influencing either the transition to the naïve state or naïve cell stability. It remains to be deduced whether DNMT3A/DNMT3B impact the naïve state through the hypermethylation they contribute to or through a non-catalytic role. Additionally, in order to better understand the functional impact of resetting-associated hypermethylation, its potential impact on the differentiation potential of the naïve cells remains to be investigated independently of the known roles of de novo methyltransferases in stem cell differentiation.

Our data point towards the transcription factor network established upon resetting playing a role in the targeting or recruitment of DNMT3A to loci that gain methylation. Whilst we cannot currently differentiate between a direct interaction of DNMT3A with transcription factors or an indirect network-driven effect on targeting of the enzyme, loci-specific recruitment of DNMT3A via transcription factors has been previously demonstrated[49] and in vitro data support the ability of DNMT3A to interact directly with numerous transcription factors[50]. Our data are also indicative of the overexpressed NANOG and KLF2 coordinating de novo methylation, however studies have shown that KLF2 is not expressed in vivo in the human inner cell mass[51,52], where we also observe hypermethylation. Additionally, we observe comparable hypermethylation in naïve hESCs

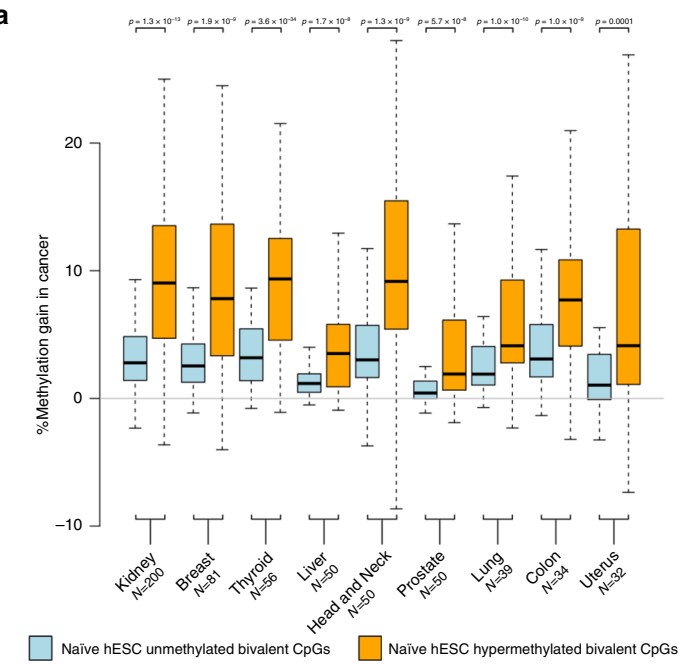

**a**

**b** Overlap Of CpGs hypermethylated in cancer with hypermethylated bivalent hESC CpGs
(CpGs tested = 303,616, bivalent CpGs hypermethylated In hESCs = 26,180)

| Cancer type | Hypermethylated in cancer | Hypermethylated in cancer and in bivalent hESCs | Fold change | p value |
|---|---|---|---|---|
| BRCA | 4494 | 980 | 7.79 | $<2 \times 10^{-16}$ |
| COAD | 1848 | 411 | 7.57 | $<2 \times 10^{-16}$ |
| HNSC | 2694 | 605 | 8.11 | $<2 \times 10^{-16}$ |
| KIRC | 10,919 | 2487 | 8.56 | $<2 \times 10^{-16}$ |
| LIHC | 2708 | 640 | 8.75 | $<2 \times 10^{-16}$ |
| LUSC | 2102 | 462 | 7.94 | $<2 \times 10^{-16}$ |
| PRAD | 2696 | 611 | 7.99 | $<2 \times 10^{-16}$ |
| THCA | 3080 | 705 | 8.33 | $<2 \times 10^{-16}$ |
| UCEC | 1682 | 373 | 7.15 | $<2 \times 10^{-16}$ |
| All cancers | 49,198 | 8568 | 11.13 | $<2 \times 10^{-16}$ |

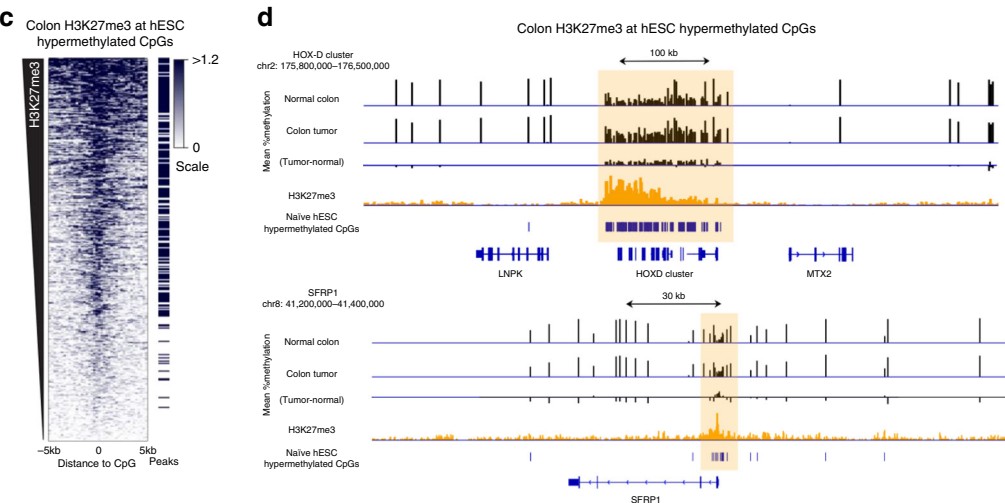

**c** Colon H3K27me3 at hESC hypermethylated CpGs

**d** Colon H3K27me3 at hESC hypermethylated CpGs

generated using two transgene-independent methods of resetting. This collectively suggests that the core pluripotency network, to which NANOG belongs, is likely responsible for coordinating the transcriptional changes that drive DNA hypermethylation. There is growing evidence in the literature regarding the acquisition of stem-like properties and expression of pluripotency genes in cancers[10,39]. This makes it intriguing to speculate that a transcriptional programme associated with the naïve pluripotency network could drive a shared mechanism of hypermethylation during resetting of primed hESCs to naïve pluripotency and in cancer development, either preceding or in conjunction with genetic mutations.

Hypermethylation during primed to naïve resetting affects developmental genes whose expression is generally low and is

**Fig. 7 Resetting-associated hypermethylation is mirrored in cancer. a** Differences in mean methylation level between matched normal tissue and tumour samples of bivalent CpGs identified as hypermethylated ($N = 23,123$) or not hypermethylated ($N = 25,977$) during the transition to the naïve state in hESCs. Data are presented for 592 individuals, separated by tumour location. $P$-values determined via paired Wilcoxon test (two-sided). CpGs used for analysis were filtered for those that are unmethylated in primed hESCs ($\beta < 0.3$). Lines=median; Box=25th–75th percentile; whiskers=1.5× interquartile range from box. **b** Overlap of CpGs hypermethylated in cancers with bivalent CpGs hypermethylated during hESC resetting. Values for all cancers were generated by testing all cancer samples against all normal samples. $P$-values are determined using Fisher's exact tests (two-sided). **c** Heatmap of H3K27me3 distribution in 250 bp windows around each hESC Hypermethylated CpG in normal colon ($N = 26180$), ordered by total H3K27me3 reads within 5 kb. Peaks are taken from ENCODE (Supplementary Table 8). Scale is in normalised reads per million. **d** Selected genomic regions containing hESC Hypermethylated CpGs marked by H3K27me3 in the normal colon (highlighted light orange). Mean methylation in normal colon and colorectal tumours from TCGA, as well as the difference between them are shown in black. hESC Hypermethylated CpGs are shown in blue, and H3K27me3 ChIP reads from ENCODE are shown in orange (Supplementary Table 8). Scale for methylation data is 0–100%. Scale for methylation difference is −50 to 50%. Scale for H3K27me3 is 0 to 30 normalised reads per million reads.

further attenuated upon hypermethylation, as is often observed in cancer[33,38]. The function of hypermethylation in cancer remains a topic of debate. While several studies have shown clear repressive roles of hypermethylation for individual tumour suppressor genes[7,53], it remains less well understood what the purpose of hypermethylation of a large number of loci might be. It has been proposed that aberrant hypermethylation in cancer may act to block cellular differentiation, thus enabling cancer cells to continue to propagate in their more primitive states[33,44,54], and this has been experimentally demonstrated in a recent study[9]. During reprogramming of somatic cells to induced pluripotent stem cells (iPS), global DNA demethylation occurs late[55] and is a bottleneck for efficient reprogramming[56]. Our data indicate that in addition to global DNA demethylation which is efficiently erased in further naïve resetting, gain of DNA methylation in bivalent developmental gene promoters will lock cells in a primitive state. The commonality in methylation patterns across cancer types, each harbouring different driver mutations, suggests that these methylation changes may by regulated by a common overarching mechanism and occur early in tumourigenesis, as has been demonstrated previously in one of the few models of early cancer development[57]. In line with this, the notion that cancer cells follow an evolutionary trajectory towards a stem cell state[10,39] makes the transition from primed to naïve pluripotency an interesting model to study biological processes such as DNA methylation that likely occur early during cancer initiation, and may be analogous to dedifferentiation. Additional molecular features of the primed to naïve state transition appear analogous to cancer hallmarks[58], such as altered metabolism[12], loss of imprints[15], loss of DNA hydroxymethylation[59] and genomic instability[11,15,60]. Whether they are related to the changing epigenetic landscape remains unexplored, but further use of this model system may shed light on the emergence of these characteristics during cellular transformation. We propose that naïve resetting may provide a good model system to understand whether other molecular processes associated with cellular reprogramming play a role in tumourigenesis.

## Methods

**Cell lines**. WA09/H9 NK2 primed hESCs were kindly provided by Austin Smith[12] with permission from WiCell. All hESCs were cultured on irradiated mouse embryonic fibroblasts (iMEF). iMEFs were seeded at a density of $1 \times 10^6$ cells per 6-well plate, in 5% $O_2$, 7% $CO_2$ at 37 °C in a humidified incubator.

**Cell culture**. Primed H9-NK2 cells containing doxycycline-inducible *KLF2* and *NANOG* transgenes coupled to Venus were maintained in conventional medium (KSR/FGF) comprised of DMEM/F-12 (Sigma Aldrich) with 20% KSR (Thermo-Fisher Scientific) and 10 ng per ml basic fibroblast growth factor (bFGF; Peprotech), supplemented with 2mM L-glutamine (ThermoFisher Scientific), 100 μM 2-mercaptoethanol (2ME) (ThermoFisher Scientific), 1% MEM non-essential amino acids (ThermoFisher Scientific), and 50 mg per ml Penicillin-Streptomycin. Cultures were passaged every 5–6 days as small clumps by dissociation with a buffer containing 1 mg per ml Collagenase IV (ThermoFisher Scientific), 0.025% Trypsin

(ThermoFisher Scientific), 1 mM $CaCl_2$ and KSR at a final concentration of 20% in PBS. Medium was changed daily.

Resetting to the naïve state was carried out as previously described[12]. Conventional hESCs were dissociated to single cells with trypsin and re-plated in the presence of 10 μM Rho-associated kinase inhibitor (ROCKi [Y-27632]; Sigma Aldrich). After 24 h, media was changed to primed media with 1 μM doxycycline (Sigma Alrich). The following day, media was changed to 2iL+dox media composed of 50% DMEM/F12 and 50% Neurobasal (ThermoFisher Scientific) supplemented with 2mM L-glutamine, 100 μM 2ME, N2 (ThermoFisher Scientific), B27 (ThermoFisher Scientific), 1 μM PD0325901 (StemCell Technologies), 1 μM CHIR99021 (StemCell Technologies), human recombinant LIF (Peprotech), 1x Penicillin-Streptomycin and 1 μM doxycycline. Media was changed daily. Cells were split every 4–5 days after dissociation to single cells using Accutase (Sigma Aldrich). After 2 weeks, doxycycline was withdrawn and PKC inhibitor Gö6983 (Sigma Aldrich) was added at a concentration of 5 μM. Cells in 2iL+Gö were split every 4–5 days after dissociation to single cells using Accutase.

**Stable knock down or overexpression cell line generation**. Short hairpin RNA (shRNA) constructs were obtained from Dharmacon in the TRC pLKO.1 lentiviral vector. Sequences are listed in Supplementary Table 1. Knockdown of DNMT3A2 was not carried out due to an inability to design an effective shRNA against its single unique exon. Entry clone for overexpression of TET1 was obtained from Harvard PlasmID Repository (TET1 in pENTR223; HsCD00399189) and a recombination reaction was performed with the pLenti CMV puro DEST (w118-1) destination vector with Gateway LR clonase II (ThermoFisher Scientific) to generate expression vectors. To generate lentiviral particles, HEK293T cells were transfected with the shRNA plasmid or expression vector for a target gene, the packaging construct pCMV Δ8.91, and a vesicular stomatitis virus glycoprotein (VSV-G) containing envelope expressing plasmid pMD2.G, using jetPrime (Polyplus) at a ratio of 1:2. Primed hESCs were treated with 6 μg per ml polybrene (Sigma Aldrich), transduced with filtered lentiviral particles, and stable hESC knock down or overexpression cell lines were generated by puromycin selection (1 μg per ml) of successful integrants.

**Western blotting**. Whole cell lysates were extracted in RIPA buffer (Sigma Aldrich), with protease inhibitor cocktail (Sigma Aldrich). Proteins (concentration determined by BCA assay) were separated by electrophoresis on a 4–12% Bis–Tris gel in MOPS running buffer (ThermoFisher Scientific) and then transferred to polyvinylidene difluoride (PVDF) membranes (Merck Millipore). Membranes were blocked with 5% skimmed milk for 45 min at room temperature and incubated overnight at 4 °C with primary antibody (TET1: Source Bioscience; GTX124207 at 1:1000 and GAPDH: Cell Signalling Technologies; 2118S at 1:2500) in blocking buffer. Membranes were incubated for 1 h at room temperature with horseradish peroxidase-conjugated secondary antibodies sheep-anti-mouse IgG or sheep-anti-rabbit IgG (1:5,000; GE Healthcare; NA931, NA934). Membranes were washed in 0.1% Tween-20 in PBS (PBST), and detection was performed with enhanced chemiluminescence (ThermoFisher Scientific), with visualisation on the Amersham Imager 600 (GE Healthcare). Uncropped blots are provided in Supplementary Fig. 11.

**qPCR**. Cells were dissociated to single cells using Accutase and serially plated for 2 h to eliminate excess iMEFs. Total RNA was isolated from pelleted hESCs using the Direct-zol RNA mini-prep kit (Zymo) and treated with the DNA-free™ DNA removal kit (ThermoFisher Scientific). Complementary DNA (cDNA) was made using a high-capacity RNA to cDNA kit (ThermoFisher Scientific). Real-time PCR was carried out using one-step Sybr green reaction mix (Bio-Rad) on the CFX384 Touch™ Real Time PCR detection system (Bio-Rad). An endogenous control (GAPDH) was used to normalise expression. Primer sequences are listed in Supplementary Table 2.

**Chromatin immunoprecipitation**. Cells were cross-linked with 1% formaldehyde for 10 min at room temperature with gentle rocking, after which the formaldehyde

was quenched with 1.25 M glycine. Chromatin was then extracted from the cross-linked cells using the chromatin extraction kit (Abcam), as per the manufacturer's instructions. The extracted chromatin was then fractionated by sonication at 4° (12 cycles of 15 s on, 60 s off; Diagenode Bioruptor® Plus). The size of the sonicated chromatin was then checked by agarose gel electrophoresis. Chromatin immuno-precipitation was carried out using the ChIP—One Step kit (Abcam) with a starting total of 5 μg of chromatin. Immunoprecipitation was carried out as per the man-ufacturer's instructions and the following quantites of antibody were used for immunoprecipitation; H3K4me3 0.5μg (Abcam; ab8580), H3K27me3 2 μg (Abcam; ab195477). As a loading control for assessing immunoprecipitation we isolated input DNA for each sample, which represent the starting quantity of chromatin prior to immunoprecipitation. Input and immunoprecipitated DNA were quanti-fied by real-time PCR, and data are shown as the % enrichment relative to the input for each sample. Primer sequences are listed in Table 3.

**Bisulfite Sequencing Analysis**. Bismark coverage files downloaded from GEO were uploaded into SeqMonk (v1.41.0), where the genomes were binned into 300 bp probe windows. Methylation quantitation was carried out using the 'Bisulphite methylation over features' pipeline in SeqMonk (v1.41.0), with a 300 bp probe carried forward if it contained at least 5 CpGs each with at least 3 counts. Motif enrichment analysis was performed using the analysis of motif enrichment (AME) tool on the MEME suite (v5.0.4)[61], searching against the human HOCOMOCO (v11 FULL) database. Sequences were scored using the average odds score and motif enrichment calculated using Fisher's exact test.

**Overlap analysis**. Overlap analysis was performed in R using the package regi-oneR (Version 3.8: https://bioconductor.org/packages/release/bioc/html/regioneR. html). Overlap was performed using the 'overlapPermTest' function with 1000 permutations. Random regions were generated for the hg19 genome using the 'circularRandomizeRegions' function. Random loci generation was restricted to loci present in the Illumina EPIC array (for overlaps performed with Illumina EPIC array probes) or to regions with a (G + C) fraction >0.55 and a CpG observed-to-expected ratio >0.6 (for overlaps performed with bisulfite sequencing data). ENCODE and ChromHMM data for the H1 hESC cell line were downloaded from the UCSC genome browser. For ENCODE data, StdPk files were downloaded for each histone modification and genomic coordinates extracted (as BED files) for use in the overlap analysis.

**Mass spectrometry of nucleosides**. Genomic DNA was digested using DNA Degradase Plus (Zymo Research) according to the manufacturer's instructions and nucleosides were analysed by LC-MS/MS on a Q-Exactive mass spectrometer (Thermo Scientific) fitted with a nanoelectrospray ion-source (Proxeon). All samples and standards had a heavy isotope-labelled nucleoside mix added prior to mass spectral analysis (2′-deoxycytidine-$^{13}C_1$, $^{15}N_2$ (Santa Cruz), 5-(methyl-$^2H_3$)-2′-deoxycytidine (Santa Cruz), 5-(hydroxymethyl)-2′-deoxycytidine-$^2H_3$ (Toronto Research Chemicals). MS2 data for 5hmC, 5mC and C were acquired with both the endogenous and corresponding heavy-labelled nucleoside parent ions simulta-neously selected for fragmentation using a 5 Th isolation window with a 1.5 Th offset. Parent ions were fragmented by Higher-energy Collisional Dissociation (HCD) with a relative collision energy of 10%, and a resolution setting of 70,000 for MS2 spectra. Peak areas from extracted ion chromatograms of the relevant frag-ment ions, relative to their corresponding heavy isotope-labelled internal standards, were quantified against a six-point serial 2-fold dilution calibration curve, with triplicate runs for all samples and standards.

**Targeted bisulfite sequencing**. Bisulfite PCR primers were designed against an in silico bisulfite converted reference sequence using the Bisulfite Primer Seeker software (Zymo) or Methprimer (Urogene), and universal Illumina adapter sequences were added to the 5′ end of each primer. Cells were dissociated to single cells using Accutase and serially plated for 2 h to eliminate excess iMEFs. DNA was isolated from pelleted cells using the PureLink Genomic DNA mini kit (Ther-moFisher Scientific). Bisulfite conversion of DNA was carried out using the Imprint® DNA Modification kit (Sigma Aldrich), following the manufacturer's instructions. The modified DNA was amplified using the loci-specific bisulfite PCR primers (listed in Supplementary Table 4) and HotStar Taq DNA Polymerase (Qiagen). The PCR conditions were as follows: 95 °C for 15 min; 94 °C for 30 seconds; 56 °C for 30 s; 72 °C for 1 min; Repeat steps 2–4 29×; 72 °C for 10 min; Hold 12 °C. PCR products were purified using SPRI beads (Agencourt AMPure XP, Beckman Coulter). Amplicons were PCR amplified with 8 cycles using a universal Illumina forward primer and an indexed reverse primer and quantified with the Kapa Library quantification kit for Illumina (Roche). For larger experiments, multiplex targeted bisulfite sequencing was performed using the 48×48 layout on the Fluidigm C1 system (Fluidigm), coupled with Illumina MiSeq sequencing. Fluidigm primers are listed in Supplementary Table 5. Amplicons from a single sample were pooled and sequencing was performed on an Illumina MiSeq with 150 bp paired-end reads, using v3 chemistry, at Barts and the London Genome Centre (London, UK). Reads were quality trimmed and mapped to a personalised human genome composed of amplicon sequences, using Bismark (v.0.19.0), fol-lowed by extraction of methylation calls.

**Infinium MethylationEPIC BeadChip assay**. Genomic DNA was extracted using the PureLink Genomic DNA mini kit (ThermoFisher Scientific). Bisulfite con-version of DNA was carried out using the Imprint® DNA Modification kit (Sigma Aldrich), following the manufacturer's instructions. Infinium MethylationEPIC BeadChip assay (Illumina) was performed according to manufacturer instructions by Barts and the London Genome Centre (London, UK). The Bioconductor package ChAMP (version 2.11.3: https://bioconductor.org/packages/release/bioc/html/ChAMP.html) was used to process raw Infinium idat files using the GRCh37 human genome manifest file.

**TCGA analysis**. Illumina 450K DNA Methylation data spanning 396965 CpGs and 9664 samples was downloaded from the Pan Cancer Atlas (https://gdc.cancer.gov/about-data/publications/pancanatlas). All samples from individuals without both a tumour and normal tissue sample were removed. Samples from tumour types with less than 30 individuals were removed. In order to assess only CpGs deemed "bivalent", CpGs outside of regions that showed a peak of H3K27me3 and H3K4me3 in ENCODE H1 hESCs were removed. For this analysis, raw infinium IDAT files from the hESC resetting experiment were processed using minfi and normalised via the single-sample Noob method[62]. CpGs used for analysis were filtered for those that are unmethylated in primed hESCs (mean beta < 0.3). Unmethylated probes were restricted to those CpGs with mean Beta < 0.3 during the primed to naïve transition. Hypermethylated hESC probes were defined using ChAMP and restricted to those CpGs with ΔBeta > 0.1 in either the early transi-tion, late transition or naïve state. Probes hypermethylated in cancer were also defined using ChAMP, and similarly restricted to those CpGs with ΔBeta > 0.1 between normal tissue and tumour samples. The overlap enrichment of cancer hypermethylated and hESC hypermethylated regions was determined via Fisher's exact test. For the creation of heatmaps, data were first ordered by sample based on mean methylation of all CpGs, and then by CpG based on mean methylation across all samples of every cancer type. Statistical significance was calculated using a paired Wilcoxon test.

**Human tissue H3K27me3 analysis**. H3K27me3 and control read alignments were downloaded as BAM files for each normal human tissue examined from ENCODE (https://www.encodeproject.org/)[63,64] that corresponded to the cancers profiled by TCGA, except HSNC for which no obvious corresponding normal sample was available. Bigwig files for genome browser visualisation and peaks of H3K27me3 for comparison were obtained in the same way. ENCODE IDs for each experiment and data file can be found in Supplementary Data 1. To examine H3K27me3 levels around naïve hypermethylated CpGs, windows of 500 bp was defined centred around each CpG. ChIP-seq read counts/window were calculated using BEDtools' coverage function. Read counts were scaled to counts per 10 million based on total number of mapped reads/sample and divided by the input read count to provide a normalised read count. To prevent windows with zero reads in the input sample generating a normalised count of infinity, an offset of 0.5 was added to all windows prior to scaling and input normalisation. Regions where coverage was 0 in all samples were removed from the analysis. A similar procedure was used to generate heatmaps of H3K27me3 levels, using multiple 250 bp windows to span 5 kb on either side of each CpG. Colour scales for ChIP-seq heatmaps range from the minimum to the 90% quantile of the normalised read count.

**Analysis of 5hmC by glucMS-qPCR**. Genomic DNA was treated with T4 Phage β-glucosyltransferase (T4-BGT; NEB) according to the manufacturer's instructions. Glucosylated genomic DNA was digested with 10 U of either HpaII, MspI or no enzyme (mock digestion) at 37 °C overnight, followed by inactivation for 20 min at 80 °C. The HpaII- and MspI-resistant fraction was quantified by qPCR using primers designed around at least one HpaII/MspI site, and normalising to the mock digestion control. Resistance to MspI directly translates into percentage of 5hmC, whereas 5mC levels were obtained by subtracting the 5hmC contribution from the total HpaII resistance. Primers used are listed in Supplementary Table 6.

**Mass spectrometry-based proteomics**. Cells from three independent biological replicates per condition were washed twice with ice cold PBS supplemented with 1 mM $Na_3VO_4$ and 1 mM NaF, lysed in urea buffer (8 M urea in 20 mM in HEPES pH 8.0, 1 mM $Na_3VO_4$, 1 mM NaF, 1 mM $Na_4P_2O_7$ and 1 mM sodium β-glycer-ophosphate) for 30 min and homogenised by sonication (15 cycles of 30 s on 30 s off; Diagenode Bioruptor® Plus). Insoluble material was removed by centrifugation at 20,000g and protein levels in the cell extracts were quantified by bicinchoninic acid (BCA) analysis. For trypsin digestion, 100 μg of protein was reduced and alkylated by sequential incubation with 10 mM DTT and 16.6 mM iodoacetamyde for 1 h and 30 min, respectively. Urea concentration was diluted to 2 M with 20 mM HEPES (pH 8.0), 80 μL of preconditioned trypsin beads [(50% slurry of TLCK-trypsin (Thermo-Fisher Scientific; Cat. #20230)] were added and samples were incubated for 16 h at 37 °C with agitation. Peptide solutions were desalted using 10 mg OASIS-HLB cartridges (Waters, Manchester, UK). Briefly, OASIS cartridges were accommodated in a vacuum manifold (−5 mmHg), activated with 1 mL ACN and equilibrated with 1.5 mL washing solution (1% ACN, 0.1% TFA). Peptides were loaded into the cartridges, washed with 1 mL of washing solution, eluted with 500 μL of ACN solution (30% ACN, 0.1% TFA), dried in a speed vac

(RVC 2-25, Martin Christ Gefriertrocknungsanlagen) and stored at −80 °C. Dried peptides were dissolved in 0.1% TFA and analysed by nanoflow ultimate 3000 RSL nano instrument coupled to a Q Exactive plus mass spectrometer (Thermo Fisher Scientific). Gradient elution was from 3 to 35% solvent B in 120 min at a flow rate 300 nL/min with solvent A being used to balance the mobile phase (buffer A was 0.1% formic acid in water and B was 0.1% formic acid in acetonitrile). The spray voltage was 1.95 kV and the capillary temperature was set to 255 °C. The Q-Exactive plus was operated in data dependent mode with one survey MS scan followed by 15 MS/MS scans. The full scans were acquired in the mass analyser at 375–1500 $m/z$ with the resolution of 70,000 and the MS/MS scans were obtained with a resolution of 17,500. Overall duty cycle generated chromatographic peaks of approximately 30 s at the base, which allowed the construction of extracted ion chromatograms (XICs) with at least 10 data points. Mascot Daemon 2.5.0 was used to automate peptide identification from MS data. Peak list files (MGFs) from RAW data were generated with Mascot Distiller v2.5.1 and loaded into the Mascot search engine (v2.5) in order to match MS/MS data to peptides. Searches were performed against the SwissProt Database (release December 2015) with a FDR of ~1% and restricted to the human entries. Mass tolerance of ±10 ppm for the MS scans and ±25 mmu for the MS/MS scans, 2 trypsin missed cleavages, carbamidomethyl Cys as a fixed modification and PyroGlu on N-terminal Gln and oxidation of Met as variable modifications were allowed. The in-house developed Pescal software was used for label-free peptide quantification as described before[65]. XICs for all the peptides identified across all samples were constructed with ±2 min and ±7 ppm retention time and mass windows, respectively. Peak areas from all XICs were calculated. The maximum intensity value for the two technical replicates was selected and used for further analysis. Intensity values for each peptide were normalised to total sample intensity. Statistical significance was calculated using two tail unpaired Student's t test. Multiplicity correction was performed by applying the Benjamini–Hochberg method on the p-values, to control the false discovery rate (FDR). Differences were considered significant when FDR < 0.05. Proteins with a Mascot score > 40 were used for analysis. Data are available in Supplementary Data 2.

**RNA-sequencing**. Total RNA was extracted using Direct-zol RNA mini kit (Zymo) and DNase treated (ThermoFisher Scientific), before mRNA was isolated from 500 ng of total RNA using Dynabeads mRNA DIRECT purification kit (ThermoFisher Scientific) and fragmented with RNA fragmentation reagent (ThermoFisher Scientific). First strand cDNA synthesis was performed with SuperScript III First-Strand Synthesis System and 3 µg µl$^{-1}$ random hexamers (ThermoFisher Scientific) followed by second strand synthesis with DNA polymerase I and RNase H. After purification using SPRI beads, the double stranded cDNA was ligated to in house designed adapters (based on TruSeq Indexed adapters (Illumina)) using NEBNext Ultra II (NEB) followed by 15 cycles of amplification and library purification. Library size distribution and molarity was assessed by the DNA 1000 assay on the 2100 Bioanalyzer (Agilent), and libraries were quantified with the Kapa Library quantification kit for Illumina (Roche). Sequencing was performed on an Illumina NextSeq with 75 bp paired-end sequencing at Barts and the London Genome Centre (London, UK). Read quality was determined using FASTQC. Genomic mapping of short reads was performed using hisat2 (v. 2.1.0) to the human genome (GRCh38). Reads were counted for each sample using FeatureCounts (Subread, v. 1.6.3)[66]. RNA-sequencing analysis was performed using the R package EdgeR (v3.18.1)[67]. Upregulated and downregulated genes were called as those with Benjamini-Hochberg corrected FDR < 0.05 and a log2 fold change > 1. Pathway enrichment analysis was performed using DAVID Bioinformatics Resources[68,69].

**Alkaline phosphatase assay**. Cells were seeded a 96-well plate. After 24 h, the Amplite™ Colorimetric Alkaline Phosphatase Assay kit (Stratech) was used to measure alkaline phosphatase activity according to manufacturer's instructions.

**Flow cytometry, fluorescence-activated cell sorting (FACS)**. Cells were dissociated to single cells with Accutase and washed with 3%FCS/PBS, before being blocked in 10% FCS/PBS. Cells were resuspended in 2.5ul SUSD2-PE antibody (Biolegend; 327406) and 5ul anti-feeder-APC antibody (Miltenyi Biotec; 130-120-802) for 15 min at 4 °C in the dark. Alternatively, following 1 h pulse labelling with 10um BrdU, cells were fixed, permeabilised, blocked and stained using the APC BrdU Flow Kit (BD Pharminogen) following manufacturer's instructions, with the addition of 5ul of anti-feeder-PE antibody (Miltenyi Biotec; 130-120-166). Cells were wash twice in 3% FCS/PBS and then stained with DAPI for 15 min at 4 °C in the dark. Samples were either analysed on an LSR Fortessa cell analyser (BD Biosciences) or FACS sorted on the BD FACS Aria Fusion cell sorter. Flow cytometry data analysis was carried out using FlowJo Version 10 software.

**Statistical analysis**. Significance testing was performed using Prism (v.7.04, v.8.4.2) and Student's t test, one-way ANOVA or two-way ANOVA with Bonferroni post hoc tests as specified in the figure legends. Where applicable, data are plotted as mean ± SEM. Representative data are shown where experiments were repeated at least twice with similar results.

**Reporting summary**. Further information on research design is available in the Nature Research Reporting Summary linked to this article.

## Data availability
The data that support this work are available from the corresponding author upon reasonable request. The mass spectrometry data have been deposited to the ProteomeXchange Consortium via the PRIDE partner repository with the dataset identifier PXD019893. The sequencing data have been deposited in the Gene Expression Omnibus under GSE128130. Additional data used include ENCODE, ChromHMM and TCGA pan-cancer data, HOCOMOCO (v11 FULL), SwissProt (Dec 2015 release). Data for human naive resetting methods were obtained from GSE60945, GSE76970, GSE90168, data for mouse 2i and serum ESCs from GSE42923, and data for human and mouse in vivo development from GSE34864 and GSE49828. Source data are provided with this paper.

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

## Acknowledgements

The authors would like to thank Prof. Austin Smith at the WT–MRC Cambridge Stem Cell Institute for providing H9 NK2 cells. We thank Dr. Charles Mein and team from Barts and The London Genome Centre for the Infinium Methylation array and Sequencing services, and we thank Dr. Vinothini Rajeeve, Dr Pedro Casado and Dr. Arran Dokal for the Mass Spectrometry Proteomics service. We thank Dr. Miguel Branco and Dr. Emily Saunderson for their advice on preparation of the manuscript. This work was supported by MRC grant MR/M01892X/1; H.P. is supported by an MRC studentship [Ref: 1650326] and The Greg Wolf fund. D.S. is a Cancer Research UK Career Development fellow (Ref: C47648/A20837), and work in his laboratory is also supported by a MRC university grant to the MRC Human Genetics Unit.

## Author contributions

Project was conceived by G.F. and developed by H.P. and M.D.R. Design of experimental work, bioinformatics analysis and data interpretation was performed by H.P. and M.D.R. Analysis of TCGA pan-cancer data was performed by J.H. Proteomics and analysis of proteomics data were performed by S.A.T. Mass spectrometry of nucleosides was performed by D.O. Research was supervised by G.F., D.S., and P.C. Manuscript was written by H.P., M.D.R., and G.F. All authors contributed to the editing of the manuscript and approved its final version.

## Competing interests

The authors declare no competing interests.
