## [Peer Review File · Nature Communications]

Reviewers' Comments:

Reviewer #1:

Remarks to the Author:

The authors show that prime to naive transition in human ES cells is associated with methylation of CpG islands for a group of developmental genes that are also methylated in cancers. The authors suggest this hypermethylation is potentially mediated by DNMT3A. The authors focus on transcription factors that are upregulated during early transition, and show by functional knockdown of factors that are upregulated that these factors may be involved in hypermethylation.

The paper is very well communicated and the results are potentially interesting. A major interesting observation is that during the widespread hypomethylation associated with reprogramming to naive state, there is a group of genes getting hypermethylated. Molecular mechanisms that cause this hypermethylation, which the authors indicate, needs further investigation to link them conclusively to the hypermethylation involved during the whole transition from early, late and final naive state. Some of these concerns are elaborated in the comments below.

- The model provides a good system to study if hypermethylation of these genes has functional consequences, which should be explored.

- It has been shown recently (Zhou et al. 2018, PMID: 29610480) that hypomethylation is linked to mitotic cell divisions, and within this context few regions get hypermethylated. The data in the current manuscript is interesting as the intermediate early and late transition stages involve rapid (is that correct?) hypermethylation of many promoters followed by demethylation. These dynamics are very interesting, but the experimental designs do not address these dynamics through which the cells seem to traverse to the final naive state wherein fewer number of these CpG sites/regions ultimately. In the context of the Zhou et al model, is there a relation to mitotic cell divisions and appearance of the methylation changes?

- Fig. 1b: How do early and late transition look for the majority of probes that get demethylated during primed to naive reprogramming? What are the numbers of CpGs acquiring methylation? How many of hypermethylated sites are in CpG-islands, non-CpG-islands?

- Fig. 1c: Does it suggest that during late transition these genes are hypermethylated and upregulated, but then again get downregulated and come to the same expression level as primed cells upon final transition to naive cells? This is counter-intuitive, and goes against the mechanism that the authors propose. Like the authors suggest, could it be 5hmC increase in the late transition state, which then may acquire 5mC. Given this confounding observation, it becomes very important to map 5hmC to understand the biology.

- Since different CpG probes are being considered in S4D and 3A, the effect of the KD cannot be easily discerned. Are there locus specific effects? Combined analyses/presentation (side by side) of the methylation changes at the different categories of differentially methylated CpGs (from early, late, naive vs. primed comparisons) in the different cell states (early, late, naive, primed) with the KDs will be required to understand role of each epigenetic enzyme, the state at which they active, and the locus affected. Currently there is little to make out from the way the data is analyzed (different probes are being analyzed in different figures).

- The magnitude of the impact of DNMT3 KDs on naive state measured by alkaline phosphatase activity, although significant, is difficult to interpret if it alone is sufficient to monitor actual defect in reprogramming to naive state. For example, are differentiation and teratoma phenotypes altered? What happens to gene expression patterns between naive (with DNMT3A KD) and control naive cells?

- In Fig. 3B, of the two different empty vector and TET1 samples (y-axis) correspond to early and primed? Is the data coming from the EPIC array? A previous study has shown that TET1 activity increases and DNMT3A activity decreases during primed → naive transition (PMID: 28457889). The authors here suggest the opposite. Are these species differences some basic difference in the underlying epigenetic mechanisms? In regard if the TETs have any role, the down regulation of TET does not fit the observation that the naive cells globally have hypomethylation (even though there are some CpG islands that get hypermethylated). So then might there be a role for other TETs? For conclusive evidence for any role of TET1 (and other TET) requires studying TET1 overexpression/KD through early, late to naive states. Further, to test if TET1 overexpression has any impact, it needs to be tested if TET1 overexpression causes hypomethylation in the primed cells.

- Throughout the study, different definitions of hypermethylation (10%, 5%) used. Firstly, these differences are very marginal (especially 5%). Secondly, is it not better to use the same definition for consistency? From the heat plots, the beta-values seem very high for the regions analyzed. So why use such low differences?

- The data for role of TFs in modulating hypermethylation is very interesting, and has been proposed by multiple studies (pioneered by Schübeler group, Gebhard et al. PMID: 20145141). The authors attribute TFs for directing hypermethylation in the said CpG sites/regions. However, to rule out that these findings could be incidental, occupancy of the target regions by the TFs is required in control and the KD conditions for direct and conclusive evidence for their role in hypermethylation. Moreover, TF binding is generally believed to protect from DNA methylation (Schübeler group), however the data here suggest that these TFs are required for hypermethylation. Are the authors missing TFs that are critical as they focus on TFs that are overexpressed in their model? Moreover, TFs (expression levels) has been suggested to have minor consequences opposed to their recruitment efficacies. This issue should be considered and discussed. In this case, additional analyses is required to test if the CGI promoters studied here are prevented from silencing (and increase H3K4me3).

Reviewer #2:

Remarks to the Author:

The manuscript of Patani et al. investigated the changes of DNA methylation during the resetting of primed hESCs into the naïve 2iLGo state using the NANOG/KLF2 dox-inducible transgenes. It is well known that this process results in a dramatic shift from a DNA hypermethylated primed state to a hypomethylated naïve state. The authors interrogated this process using Infinium MethylationEPIC array and mass spectrometry and focused on the hypermethylated sites. In principle this is refreshing and they were able to reveal hypermethylation patterns on the bivalent CGIs during the resetting, which is followed by identification of candidate transcription factors responsible for the dichotomy on the hypermethylation and subsequently they described that the same phenomena is also present in cancer.

Although the study is interesting, it is somehow confusing, at points the data analysis and follow-up experiments becomes convoluted or not properly performed to address the question. This in turn makes the claims less convincing and rather confusing (I have tried to point out the main problems below). In my opinion, the manuscript has potential, but not only the concerns raised below need to be addressed, but also it needs to be made clear the main messages in a coherent narrative.

I hope the following comments are of help to the authors:

The authors utilized the system and protocol established by Takashima et al., 2014, did the authors check the degree of leakiness of the NANOG/KLF2 dox-inducible transgenes as this might affect the follow-up investigation of the mechanisms of bivalent CGI hypermethylation.

In line 68-75, it is interesting that the authors looked into the hypermethylated CpGs during the resetting from primed to the naïve state, for example the HOXA cluster. It would be more informative if the authors can show the methylation of the same HOXA cluster of late transition +PKCi or -PKCi samples in figure 1c to support the claim that this is independent of PKCi. Why the authors have changed the way that they define a hypermethylated region (10% vs 5%) when whole-genome bisulfite-seq was analyzed is not clear. On this note, I believe that a correlation analysis rather than the presented enrichment analysis between the previously published data and their data set will be more informative and improve the conclusion on line 100.

The conclusion that the hypermethylated genes are "attenuated" but "are characterised by low average expression in primed hESCs (Fig. 2b)" is interesting. However, I believe that the authors need to explore this more. Showing a boxplot is not enough....please break apart more the information as you did with the Hox genes. For example are these the bivalent domains, TFs?, show several examples, can you cluster them using the transcriptional dynamics or the methylation dynamics?

On this note, the authors concluded that the HOX gene family becomes hypermethylated but are upregulated upon hESC reprogramming (Figure 2c), and the authors explained that this might be caused by the heterogeneity or 5-hydroxyMethyl-C. Would the authors be able to assess the heterogeneity of their culture since its recently reported that naïve 2iLGo culture contains an intermediate subpopulation (Messmer et al., Cell reports, 2019). Also, why the comment in regards to 5-hydroxyMethyl-C? How hypermethylation can be confused with 5OH-Methyl-C?

The rationale for the experiments in figure 3 need to be better explained or not explained at all, since KD or OE epigenetic regulators (DNMTs or TETs) are expected to affect the methylation levels. Furthermore, it is difficult to understand figure 3 and the conclusion from such experiments, e.g. why the effect of KD or OE of these regulators at the hypermethylated bivalent regions identified in figure 1 were not examined?

The author suggested that the isoform DNMT3A2 is responsible for de novo DNA methylation based on the observation that isoform DNMT3A1 knockdown did not impact on the methylation. Has the author also conducted the knockdown of the isoform DNMT3A2? It would be better to show the results in parallel with the knockdown of DNMT3A1.

The author mentioned that NANOG and KLF2 or the associated networks might be the factors coordinating the hypermethylation based on the observation that removal of MEK inhibitor or GSK3 β inhibitor during reprogramming did not impact on the hypermethylation in Figure 5d. Indeed, Was not this expected?

One of the key claims (abstract) is that NANOG is a key driver of the hypermethylation of bivalent regions but the presented results do not convincingly support this conclusion. Is there any experiment that can be done to more specifically consolidate the role of NANOG on hypermethylation in the context of primed to naïve resetting?

The main message of this manuscript is that the hypermethylated bivalent regions identified during primed to naïve resetting can be found in cancer. The authors examined the link in one figure (figure 7). Again, I found this is not convincing. Not only the analysis lacks depth but the authors did not interrogate the mechanism at all. This would greatly strength the claims of the manuscript.

Minor points:

The introductory paragraph is not convincing, in the sense that it does not lead well to why this study was designed to answer the "oncogenic" question. The sentence in line 46 is not enough. The authors might consider using 'resetting' 'reversion' 'toggling' instead of reprogramming for

primed hESCs to naïve transition to avoid confusion.

The statistical analyses is not clear at all times, neither much of the bioinformatic analysis and should include supplementary data for important analysis.

Completely a minor point: The figures are call in lower case, eg. Figure 2b, however in the real figures, the figure is call in upper cases "B"

We are pleased to know that both reviewers find our study interesting and appreciate the novelty in addressing the phenomenon of DNA hypermethylation during primed to naïve hESC reprogramming. We believe our study is essential to start understanding the mechanism of DNA hypermethylation in cancer and that our model system will pave the way for further study of other reprogramming-associated processes that may contribute to tumorigenesis.

In the attached resubmission, we have addressed the criticisms raised by the referees and provide new data to address these points.

Reviewers' comments:

Reviewer #1 (Remarks to the Author):

The authors show that prime to naïve transition in human ES cells is associated with methylation of CpG islands for a group of developmental genes that are also methylated in cancers. The authors suggest this hypermethylation is potentially mediated by DNMT3A. The authors focus on transcription factors that are upregulated during early transition, and show by functional knockdown of factors that are upregulated that these factors may be involved in hypermethylation.

The paper is very well communicated and the results are potentially interesting. A major interesting observation is that during the widespread hypomethylation associated with reprogramming to naïve state, there is a group of genes getting hypermethylated. Molecular mechanisms that cause this hypermethylation, which the authors indicate, needs further investigation to link them conclusively to the hypermethylation involved during the whole transition from early, late and final naïve state. Some of these concerns are elaborated in the comments below.

We would like to thank the reviewer for the constructive criticism. We have made significant changes to the manuscript through the addition of new data and reformulation of text where needed.

1. The model provides a good system to study if hypermethylation of these genes has functional consequences, which should be explored.

We agree that this is a good system to study the functional consequences of hypermethylation. We have extended our analysis using the single knock down cell lines for DNMT3A and DNMT3B, by performing gene expression analysis of naïve pluripotency genes. Furthermore, we generated a double knockdown of DNMT3A and DNMT3B and reprogrammed these cells to the naïve state where we measured the effect on DNA methylation when both genes are knocked down. We performed both targeted gene expression and DNA methylation analysis, and the alkaline phosphatase assay on these naïve cells to investigate any consequences of the reduced methylation we observe in the double KD cells in the naïve pluripotent state. This data has been added to Fig. 3c-g, Supplementary Fig 5e, 5f.

An interesting outcome of the double knockdown of DNMT3A and DNMT3B was that compared to the single knockdowns, the naïve cell identity is even more severely affected, suggesting a synergy between 3A and 3B (as measured by alkaline phosphatase activity). This indicates that de novo methyltransferases have significant roles in the naïve phenotype, partially compensating each other in naïve cells, and that at this stage we cannot separate between the catalytic and non-catalytic roles of DNMT3A and DNMT3B in establishing the naïve identity. This suggests that hypermethylation of

some of the promoter CGIs may be functionally linked to naïve phenotypes during the transition to naïve cells, but this experiment would need to be tested in catalytically mutant DNMT3A/DNMT3B context (so that the rest of the protein is still intact). The in depth characterisation of the role of DNMT3A/3B loss in sustaining naïve pluripotency will be done at a later stage and is outside the scope of this study.

To further explore the functional consequences of hypermethylation, we additionally used a published cell-surface marker of naïve pluripotency, SUSD2 (Bredenkamp et al, 2019 Stem Cell Reports, PMID: 31031191), to separate cells during reprogramming into SUSD2 positive and negative fractions, where the SUSD2+ fraction enriches for cells that have already acquired naïve properties and SUSD2- would be a population that do not yet display properties of naïve cells. In this experiment we tested the hypothesis that the acquisition of naïve pluripotency and hypermethylation are linked, therefore the expectation would be that the SUSD2+ population would enrich for cells that are undergoing hypermethylation and the SUSD2- population would have lower levels of hypermethylation. We carried out targeted DNA methylation analysis on these fractions and also performed gene expression analysis of naïve pluripotency genes to assess their pluripotency. While SUSD2+ cells express higher levels of naïve pluripotency genes than SUSD2- cells, the DNA hypermethylation between the two fractions is not significantly different. This means that the acquisition of naïve pluripotency and promoter CGI hypermethylation are not strictly linked, and that while hypermethylation may be necessary for the induction of naïve cell identity, it is not sufficient. This data has been added to Fig 4a-c, Supplementary Fig 7a, 7b.

Putting the above information together, we have modified the text with some further conclusions with respect to the functional consequences of hypermethylation.

2. It has been shown recently (zhou et al. 2018, PMID: 29610480) that hypomethylation is linked to mitotic cell divisions, and within this context few regions get hypermethylated. The data in the current manuscript is interesting as the intermediate early and late transition stages involve rapid (is that correct?) hypermethylation of many promoters followed by demethylation. These dynamics are very interesting, but the experimental designs do not address these dynamics through which the cells seem to traverse to the final naïve state wherein fewer number of these CpG sites/regions ultimately. In the context of the Zhou et al model, is there a relation to mitotic cell divisions and appearance of the methylation changes?

Throughout the study, where primed hESCs are used as control cells for the levels of DNA methylation, these cells have been cultured in parallel with the reprogramming cells (i.e. for the same length of time and number of passages), which was a strong indication to us that hypermethylation is not brought about simply by mitotic cell divisions as both types of cells are continuously cycling.

To further probe this, we have measured the doubling rate of primed hESCs and cells during the early transition of reprogramming and have found these not to be significantly different. Additionally, when labelled with BrdU, primed hESCs and cells in the early transition of reprogramming lose the BrdU at the same rate, indicating that the gain of hypermethylation observed at the early transition cannot be a result of mitotic division or cell selection. This data has now been added as a supplementary figure, in Supplementary Figure 1h.

3. Fig. 1b: How do early and late transition look for the majority of probes that get demethylated

during primed to naive reprogramming? What are the numbers of CpGs acquiring methylation? How many of hypomethylated sites are in CpG-islands, non-CpG-islands?

A heatmap displaying the top 20,000 hypomethylated probes has been added to Supplementary Figure 1c, demonstrating that the majority of probes are gradually demethylated during primed to naïve reprogramming. Additionally, the numbers of probes that are hypo- or hypermethylated are shown on a bar plot (Supplementary Fig. 1b), and for those that are hypermethylated, there is a further bar plot depicting the proportions of probes that reside within CpG islands, shores, shelves or outside of CpG islands (Supplementary Fig. 1d). This analysis demonstrates that the majority of CpG probes that are hypermethylated during the transition and in naïve cells compared to primed cells reside within CpG islands and shores.

4. Fig. 1c: Does it suggest that during late transition these genes are hypermethylated and upregulated, but then again get downregulated and come to the same expression level as primed cells upon final transition to naive cells? This is counter-intuitive, and goes against the mechanism that the authors propose. Like the authors suggest, could it be 5hmC increase in the late transition state, which then may acquire 5mC. Given this confounding observation, it becomes very important to map 5hmC to understand the biology.

We thank the reviewer for this observation, which is indeed counterintuitive. We have performed a gluc-MS qPCR assay to measure both 5mC and 5hmC at a selection of CpGs within the HOX genes to determine whether these genes are gaining 5hmC rather than 5mC (Figure 2e), however this does not seem to be the case during the late transition. It is therefore more likely that the observation regarding hypermethylation and upregulation of HOX genes is a result of heterogeneity within the reprogramming population of cells. This would mean that the cells that hypermethylate (estimated ~60% of the total, which is the average estimated level of hypermethylation) would inhibit expression of the HOX genes but the rest of the population would overexpress it. Unfortunately, there is no readily available method that allows us to sort cells that hypermethylate from the total population (and SUSP2 sorting did not help in this regard either) therefore we can't test this hypothesis.

5. Since different CpG probes are being considered in S4D and 3A, the effect of the KD cannot be easily discerned. Are there locus specific effects? Combined analyses/presentation (side by side) of the methylation changes at the different categories of differentially methylated CpGs (from early, late, naive vs. primed comparisons) in the different cell states (early, late, naive, primed) with the KDs will be required to understand role of each epigenetic enzyme, the state at which they active, and the locus affected. Currently there is little to make out from the way the data is analysed (different probes are being analysed in different figures).

We agree that having the analyses of different time points in different figures did not make the data easy to interpret. Combined analyses of methylation changes in primed, early and late transition hESCs are now provided both in Figure 3a (using probes that are most significantly changing during the early transition) and in Figure 3b (using probes that are most significantly changing during the late transition) to enable simpler visualization of the data to better understand the roles of each enzyme.

6. The magnitude of the impact of DNMT3 KDs on naive state measured by alkaline phosphatase activity, although significant, is difficult to interpret if it alone is sufficient to monitor actual defect in

reprogramming to naive state. For example, are differentiation and teratoma phenotypes altered? What happen to gene expression patterns between naive (with DNMT3A KD) and control naive cells?

We agree that alkaline phosphatase on its own is insufficient to determine the impact of DNMT3A or 3B KD on pluripotency. As mentioned above, we have extended this by performing gene expression analysis for naïve pluripotency genes in the KD cells as well as performing both alkaline phosphatase and gene expression analysis in DNMT3A/DNMT3B double KD cells. Our results indicate that a knock down of DNMT3A, as well as DNMT3A/DNMT3B combined results in reduced stability of naïve pluripotency as measured by alkaline phosphatase and gene expression of naïve pluripotency genes. We also see a significant reduction in hypermethylation in naïve cells in the DNMT3A/DNMT3B double KD compared to control cells (Figure 3g), indicating that hypermethylation or the *de novo* methyltransferases are important for naïve pluripotency.

We have considered performing experiments to investigate the effects of KD on differentiation, and have previously attempted to generate embryoid bodies from these cells. However, because both DNMT3A and DNMT3B proteins are required for differentiation, it is not possible to disentangle whether any effect observed is as a result of the reduction in hypermethylation in the naïve cells or due to the absence of the proteins themselves as the cells differentiate. Deducing the impact of the absence of hypermethylation on differentiation without the confounding effect of the absence of the DNMT3 proteins would require an inducible KD system that enables re-introduction of DNMT expression prior to differentiation.

7. In Fig. 3B, of the two different empty vector and TET1 samples (y-axis) correspond to early and primed? Is the data coming from the EPIC array? A previous study has shown that TET1 activity increases and DNMT3A activity decreases during primed -> naive transition (PMID: 28457889). The authors here suggest the opposite. Are these species differences some basic difference in the underlying epigenetic mechanisms? In regard if the TETs have any role, the down regulation of TET does not fit the observation that the naive cells globally have hypomethylation (even though there are some CpG islands that get hypermethylated). So then might there be a role for other TETs? For conclusive evidence for any role of TET1 (and other TET) requires studying TET1 overexpression/KD through early, late to naive states. Further, to test if TET1 overexpression has any impact, it needs to be tested if TET1 overexpression causes hypomethylation in the primed cells.

The missing label has been added to the TET1 graph, showing that the data comes from primed and early transition hESCs. The figure legend has also been amended to make it clear that the data plotted is that generated from targeted bisulfite sequencing, not from an EPIC array. These data are now in Supplementary Figs 6a-e.

The study cited (PMID: 28457889) is in the mouse naïve reprogramming system (2i cells) and indeed TET regulation is very different, possibly a reflection of species differences in terms of epigenetic regulation. Since we saw such a rapid loss of TET1 protein (already in the first 24 hours) and it has been shown in the literature that a TET triple KO in primed hESCs results in hypermethylation of bivalent CGIs (Verma et al., 2018 Nature Genetics; PMID: 29203910), we had to test the hypothesis that TET1 loss is involved in promoter CGI hypermethylation, which turned out not to be the case.

Regarding hypomethylation, the reviewer is suggesting that TET1 may have a role in hypomethylation in the human reprogramming system. Global hypomethylation is not the topic of this manuscript, but we have evidence that UHRF1 is lost (our proteomics data, Supplementary Table 7), and since it has been shown that the main reason of global demethylation in mouse is due to the impaired function of UHRF1 (von Meyenn et al 2016 Mol Cell; PMID: 27237052), we suspect that

there may be stronger parallels between mouse and human when it comes to the mechanism of global loss of DNA methylation.

8. Throughout the study, different definitions of hypermethylation (10%, 5%) used. Firstly, these differences are very marginal (especially 5%). Secondly, is it not better to use the same definition for consistency? From the heat plots, the beta-values seem very high for the regions analysed. So why use such low differences?

We agree that a consistent definition of hypermethylation is ideal and we have performed the analysis again with a 10% threshold for hypermethylation for all data types used.

9. The data for role of TFs in modulating hypermethylation is very interesting, and has been proposed by multiple studies (pioneered by Schübeler group, Gebhard et al. PMID: 20145141). The authors attribute TFs for directing hypermethylation in the said CpG sites/regions. However, to rule out that these findings could be incidental, occupancy of the target regions by the TFs is required in control and the KD conditions for direct and conclusive evidence for their role in hypermethylation. Moreover, TF binding is generally believed to protect from DNA methylation (Schübeler group), however the data here suggest that these TFs are required for hypermethylation. Are the authors missing TFs that are critical as they focus on TFs that are overexpressed in their model? Moreover, TFs (expression levels) has been suggested to have minor consequences opposed to their recruitment efficacies. This issue should be considered and discussed. In this case, additional analyses is required to test if the CGI promoters studied here are prevented from silencing (and increase H3K4me3).

Thank you for raising this point as we believe our data was not clearly explained in the first place. The data indicates that KD of one TF has an impact also on targets where no binding (of that specific TF) was predicted, i.e. NFKB1 KD had an impact on other regions where binding of Sox15 was predicted and vice-versa, and that regions, where none of the TFs were predicted to bind, were less affected. This implies that while TFs influence hypermethylation, their impact is not strictly dependent on their binding and may involve the larger network of TFs that is expressed during reprogramming acting directly or indirectly at gene promoters. This has been amended in the text of the manuscript for improved clarity.

To ensure that the reduction in hypermethylation observed upon knock down of transcription factors is not as a result of a delay in reprogramming, we measured the gene expression of naïve pluripotency genes in the knock down compared to control cells during the early transition of reprogramming and observed no difference between knock down and control. This data has been added to Supplementary Fig. 8b.

Our strategy of identifying TFs through cross-comparison of TF motifs and proteomics data means that our focus was on those TFs that increase in expression. This may indeed mean that we miss other important TFs, however we reasoned that this strategy was sufficient to demonstrate a proof of principle that may be applicable to other TFs, and is unlikely to be limited to those TFs that we chose to validate. Additionally, the expression and activity of TFs may not always be correlated, however here, as we see a phenotype upon knock down, it appears that there may be a relationship between the two in this experimental system. Finally, while the model of TF binding protecting from DNA methylation is present in the literature, alternative models supporting an active role for TFs in

driving hypermethylation have also been proposed (Fang et al., 2014 Mol Cell; PMID: 25219500 & Serra et al., 2014 Elife; PMID: 24623306).

The CGI promoters for which we show hypermethylation are not prevented from silencing through increased H3K4me3 as we show in our study examples of genes such as DLX5 and PITX1 that are hypermethylated and upregulated but undergo a reduction in their levels of H3K4me3 based on ChIP-qPCR (Fig. 5b).

Reviewer #2 (Remarks to the Author):

The manuscript of Patani et al. investigated the changes of DNA methylation during the resetting of primed hESCs into the naïve 2iLGo state using the NANOG/KLF2 dox-inducible transgenes. It is well known that this process results in a dramatic shift from a DNA hypermethylated primed state to a hypomethylated naïve state. The authors interrogated this process using Infinium MethylationEPIC array and mass spectrometry and focused in the hypermethylated sites. In principle this is refreshing and they were able to reveal hypermethylation patterns on the bivalent CGIs during the resetting, which is followed by identification of candidature transcription factors responsible for the dichotomy on the hypermethylation and subsequently they described that the same phenomena is also present in cancer.

Although the study is interesting, it is somehow confusing, at points the data analysis and follow-up experiments becomes convoluted or not properly performed to address the question. This in turn make the claims less convincing and rather confusing (I have tried to point out the main problems below). In my opinion, the manuscript has potential, but not only the concerns raised below need to be addressed, but also it needs to be made clear the main messages in a coherent narrative.

I hope the following comments are of help to the authors:

1. The authors utilized the system and protocol established by Takashima et al., 2014, did the authors check the degree of leakiness of the NANOG/KLF2 dox-inducible transgenes as this might affect the follow-up investigation of the mechanisms of bivalent CGI hypermethylation.

Hypermethylation happens very early in reprogramming of Primed cells, detected already at 72 hours after induction, and is a feature of other reprogramming methods (transgene independent) and in vivo human ICM cells, therefore we don't think that leakiness of NANOG/KLF2 (upon Dox retrieval during late transition) is a confounding factor in the process of CGI hypermethylation.

2. In line 68-75, it is interesting that the authors looked into the hypermethylated CpGs during the resetting from primed to the naïve state, for example the HOXA cluster. It would be more informative if the authors can show the methylation of the same HOXA cluster of late transition +PKCi or -PKCi samples in figure 1c to support the claim that this is independent of PKCi. Why the authors have changed the way that they define a hypermethylated region (10% vs 5%) when whole-genome bisulfite-seq was analyzed is not clear. On this note, I believe that a correlation analysis rather than the presented enrichment analysis between the previously published data and their data set will be more informative and improve the conclusion on line 100.

Figure 1c has been amended to show the methylation of the HOXA cluster in the presence and the absence of the PKCi, strengthening the claim that the hypermethylation is independent of the addition of the PKCi. We agree that a consistent definition of hypermethylation is ideal, as was also

pointed out by reviewer 1 and we have performed the analysis again with a 10% threshold for hypermethylation for all data types used. We also performed a correlation analysis between the hypermethylated probes detected in naïve versus primed hESCs in our InfiniumEPIC array data and the quantification of the equivalent regions from the published WGBS data set. This is displayed on a scatter plot and demonstrates further the reproducibility of the reprogramming-associated hypermethylation across independent experiments, methods and labs.

3. The conclusion that the hypermethylated genes are “attenuated” but “are characterised by low average expression in primed hESCs (Fig. 2b)” is interesting. However, I believe that the authors need to explore this more. Showing a boxplot is not enough....please break apart more the information as you did with the Hox genes. For example are these the bivalent domains, TFs?, show several examples, can you cluster them using the transcriptional dynamics or the methylation dynamics?

We have generated additional figures to support the previous data, showing a scatter plot of the genes that are hypermethylated and downregulated, similar analysis as we did with the HOX genes which has been added to Fig. 2c. Additionally we performed clustering analysis of the gene expression data and tried to identify any corresponding patterns in the methylation data. We also performed clustering analysis of the DNA methylation dynamics and tried to identify any corresponding patterns in the gene expression data, however there does not seem to be any correlation between the two, i.e. there are no patterns separating hypermethylated genes that are upregulated from those that are downregulated with regards to their gene expression or methylation dynamics. This data has been added to Supplementary Fig. 4d.

4. On this note, the authors concluded that the HOX gene family becomes hypermethylated but are upregulated upon hESC reprogramming (Figure 2c), and the authors explained that this might be caused by the heterogeneity or 5-hydroxyMethyl-C. Would the authors be able to assess the heterogeneity of their culture since its recently reported that naïve 2iLGo culture contains an intermediate subpopulation (Messmer et al., Cell reports, 2019). Also, why the comment in regards to 5-hydroxyMethyl-C? How hypermethylation can be confused with 5OH-Methyl-C?

5mC is a substrate for 5hmC, therefore can only be generated by oxidising 5mC which can result in demethylation (loss of 5mC) of promoters. We hypothesised that one of possible means that cells induce HOX gene expression is to release the inhibitory methylation signal. To address whether the observation that some of the HOX genes are hypermethylated but upregulated is because of the presence of 5hmC, we opted to use the glucMS-qPCR assay. This is because unlike bisulfite conversion-based methods for DNA methylation analysis, which do not differentiate between 5mC and 5hmC, glucMS-qPCR allows us to measure 5mC and 5hmC independently, at single CpG residues. We have performed gluc-MS qPCR assay to measure both 5mC and 5hmC at a selection of CpGs within the HOX genes to determine whether these genes are gaining 5hmC rather than 5mC (Figure 2e), however this does not seem to be the case during the late transition. It is therefore more likely that the observation regarding hypermethylation and upregulation of HOX genes is a result of heterogeneity within the reprogramming population of cells. Please see our answer to Reviewer #1, point 4 for further details.

To further explore the heterogeneity of the cell population during reprogramming, we stained cells during the reprogramming process with a published cell-surface marker of naïve pluripotency, SUSD2. We further separated these cells during reprogramming into SUSD2 positive and negative fractions, as a proxy for cells that appear more or less naïve-like. We carried out targeted DNA methylation analysis on these fractions and also performed gene expression analysis of naïve

pluripotency genes to assess their pluripotency. While SUSD2+ cells express higher levels of naïve pluripotency genes than SUSD2- cells, the DNA hypermethylation between the two fractions is comparable. (This data has been added to Fig 4a-c, Supplementary Fig 7a, 7b). This means that while there is heterogeneity in the expression of naïve cell markers, as well as heterogeneity in DNA hypermethylation (as most HOX regions are ~60% methylated), the two are independent of each other.

5. The rationale for the experiments in figure 3 need to be better explained or not explained at all, since KD or OE epigenetic regulators (DNMTs or TETs) are expected to affect the methylation levels. Furthermore, it is difficult to understand figure 3 and the conclusion from such experiments, e.g. why the effect of KD or OE of these regulators at the hypermethylated bivalent regions identified in figure 1 were not examined?

The author suggested that the isoform DNMT3A2 is responsible for *de novo* DNA methylation based on the observation that isoform DNMT3A1 knockdown did not impact on the methylation. Has the author also conducted the knockdown of the isoform DNMT3A2? It would be better to show the results in parallel with the knockdown of DNMT3A1.

Thank you for highlighting that this was not well explained. The text in the manuscript has been modified to improve the explanation of the rationale for the KD and OE experiments. While it is known that DNMTs are responsible for *de novo* DNA methylation generally, we were interested in which of the DNMTs was specifically involved in the context of reprogramming, in carrying out hypermethylation specifically at bivalent CGI promoters.

The hypermethylated probes identified in Figure 1b from the EPIC array performed across the time course of reprogramming are the same probes that are later plotted in Figure 3a and 3b (early transition or late transition hypermethylated probes, respectively). The figure legends for each of the figures have been updated for increased clarity. Additionally, based on the comment from reviewer 1, we have also combined analyses of methylation changes in primed, early and late transition hESCs for the DNMT3 KDs into a single figure to enable simpler visualization of the data to better understand the roles of each enzyme (Figure 3a, 3b).

Since knockdown of both isoforms of DNMT3A had a significant impact on DNA hypermethylation, and knockdown of DNMT3A1 had no impact on methylation, we think it is a strong argument that DNMT3A2 is most likely responsible for *de novo* methylation based on logical argumentation. This combined with the lack of commercial availability of shRNA that would target the only unique exon in DNMT3A2 that is not present in DNMT3A1 led us to focus our investigation on more novel aspects of the hypermethylation mechanism.

6. The author mentioned that NANOG and KLF2 or the associated networks might be the factors coordinating the hypermethylation based on the observation that removal of MEK inhibitor or GSK3 β inhibitor during reprogramming did not impact on the hypermethylation in Figure 5d. Indeed, Was not this expected?

To induce reprogramming in this system, in addition to NANOG/KLF2 overexpression, the MEK and GSK3b inhibitors are being added to the induced cells and we wanted to separate the two potentially confounding effects. There was a possibility that signal inhibition results in molecular events leading to *de novo* methylation of CGI promoters (and not because of NANOG/KLF2 overexpression) but testing this hypothesis demonstrated that none of the inhibited pathways were responsible for the hypermethylation.

7. One of the key claims (abstract) is that NANOG is a key driver of the hypermethylation of bivalent regions but the presented results do not convincingly support this conclusion. Is there any experiment that can be done to more specifically consolidate the role of NANOG on hypermethylation in the context of primed to naïve resetting?

We arrived to this conclusion based on eliminating any other confounding factor, so it's by exclusion that NANOG must be responsible. But we agree with the reviewer that this is not a direct evidence and, in this system, reprogramming is not possible without NANOG overexpression. However, "Epigenetic resetting" of Primed cells (Guo et al 2017 Stem Cell Reports) results in overexpression of NANOG in the first days and CGI hypermethylation, which is a further parallel with the Takashima resetting system, albeit other pluripotency genes are found to be overexpressed as well. Taking this into account, we have modified the text in the abstract to reduce the strength placed on NANOG as a definitive driver of hypermethylation.

8. The main message of this manuscript is that the hypermethylated bivalent regions identified during primed to naïve resetting can be found in cancer. The authors examined the link in one figure (figure 7). Again, I found this is not convincing. Not only the analysis lacks depth but the authors did not interrogate the mechanism at all. This would greatly strength the claims of the manuscript.

In order to further probe the mechanistic links between our observations in naïve cells and cancer, we have asked whether the regions which become hypermethylated in naïve hESCs are also marked by H3K27me3 in the normal tissues which give rise to tumours. This analysis of 8 tissues shows that the regions which gain methylation in naïve hESCs are not only hypermethylated in cancers arising in those tissues but are also marked by H3K27me3 in the equivalent normal tissue. This provides further evidence that the epigenetic reprogramming seen in naïve hESCs mirrors the aberrant epigenetic alterations seen in cancer. This data has been added to Figure 7b, 7c and Supplementary Figure 10b.

Minor points:

The introductory paragraph I not convincing, in the sense that it does not lead well to why this study was design to answer the "oncogenic" question. The sentence in line 46 is not enough.

The paragraph describing why this study was designed to address the question of hypermethylation in cancer has been expanded to demonstrate that the naïve hESCs have been used in the context of cancer previously; work that is now published (Avior et al., 2019 Cell Stem Cell; PMID: 31543368).

The authors might consider using 'resetting' 'reversion' 'toggling' instead of reprogramming for primed hESCs to naïve transition to avoid confusion.

We considered using 'resetting' in the title and manuscript, however, due to the reasons below we think 'reprogramming' is more appropriate.

Resetting currently refers specifically to the conversion from primed to naïve cells but the gap between 'somatic reprogramming to primed pluripotency' and 'primed to naïve resetting' is becoming narrower and methods are being developed to reprogram somatic cells directly to naïve cells (PMID: 31708477 , PMID: 30598530). As we show in our manuscript, promoter CGI hypermethylation is a feature of all methods resulting in induction of naïve pluripotency, including in vivo naïve cells from the inner cell mass. Therefore, any reprogramming protocol to human naïve

pluripotency, used in the research community, will have the associated CGI hypermethylation, which is specific to human naïve cells. 'Resetting' will therefore lose relevance once the gap is non-existent. 'Toggling' is not an accurate term because reversion to the naïve state is not reversible for genomic imprints and epigenetic memory of the tissue of origin.

We have changed the abstract slightly to specify the window of the reprogramming process analysed in this manuscript, i.e. from primed to naïve human pluripotency.

The statistical analyses is not clear at all times, neither much of the bioinformatic analysis and should include supplementary data for important analysis.

Where statistical tests have been used in the manuscript, we explain in detail the tests used and the significance thresholds in the figure legends. For larger analyses, this information can be found in the methods section.

The supplementary data will be provided as an additional file.

Completely a minor point: The figures are call in lower case, eg. Figure 2b, however in the real figures, the figure is call in upper cases "B"

This has been made consistent such that the text matches the figures and the figure legends throughout.

Reviewers' Comments:

Reviewer #1:

Remarks to the Author:

The authors have addressed various questions raised in response to the 1st submission. Some of the remaining concerns are listed below which can be addressed through discussions in the paper and figure edits where required. Barring these changes, the paper looks good for publication and brings forth novel findings and concepts.

1) The studies with DNMT3A/B KO and HOX gene 5mc/hmc studies lead to inconclusive results regarding the functional impact of hypermethylation. This is suggested in the rebuttal letter and could be discussed in the paper for the sake of proper context for interpreting and clarity to the reader.

2) The dynamic nature of the methylation, i.e. hypermethylation in late-transition followed by demethylation back to almost the primed-state levels should be enhanced further in the discussion/stated in the abstract. Because the current text conveys the message that the "reprogramming" ends up in a final hypermethylated state which is not the case. An accurate description in the abstract should read: TRANSITION to "naïve pluripotency and oncogenic transformation share common..." (Abstract). This is indeed an interesting finding in this model although the functions are not easily discernible.

3) Some figures regarding KD studies show only primed and late but not early, naive, and various combinations thereof (example Suppl. Fig. 5). Since it is a progression model, it is more useful to know the impact of the manipulation throughout the progression stages. For example, in Suppl. 5b ("qRT-PCR for the transcripts of the DNMT3 family in control and knock-down cell lines, across the period of resetting."), the data only shows primed and early and not "across the period of resetting". Important to carefully present this because in Fig. 3a and b "DNMT3A is not solely accountable for all the hypermethylation present". Was KD efficient throughout, what is the protein level changes in the KDs, etc. are then important considerations which the above mentioned representation of the data could capture. This leads to the important concern that can DNMT3 be attributed to all the methylation changes as suggested! It being the de novo catalytic enzyme, should/could be a key component. However one cannot exclude other factors. Since DNMT1 is also known to possess de novo activity, one cannot completely rule out DNMT1.

Further in Fig. 3b, late transition cells have similar hypermethylation in the DNMT3A and DNMT3B KD in contrast to the conclusion that "DNMT3B knockdown cells fail to hypermethylate to the same extent as control cells and DNMT3A knockdown cells show only a partial reduction in methylation". And the lack of hypermethylation in the DNMT3A and DNMT3B KD is confined to a set of probes that have 0.2 to 0.4 beta-values, while the sites that have gained high methylation tend to remain hyper hypermethylated even in the KDs.

The bottom line is the mechanistic links are not so clear, and it is still misleading to state that "We find that de novo methylation is dependent on DNMT3A" (abstract) given the uncertainty. The rest of the paper (the chromatin state links, overlap with cancer methylation, "a putative role of the de novo methyltransferases in stabilization of the naïve pluripotent state" etc.) stands firm without the mechanistic links. Inclusion of mechanism then should be discussed with the proper caveats. This reviewer agrees that it is not trivial to get at the mechanisms by KD/KO experiments because it affects viability/differentiation potential. etc. which are important components of the reprogramming model. Discussing the caveats will provide a proper perspective about the complexity.

4) KO of both 3A1 and 3A2 would have been a critical experiment at least to resolve the mechanistic basis of the role of the isoforms of 3A (as suggested by R2). KO of 3A1 but not 3A2 is attributed to lack of commercial shRNA targeting 3A2. Does this lack of available shRNA mean inability to design good shRNA? Whatever the case, this can be mentioned in methods.

Minor:

Fig. 6c middle panel y-axis scale: does it make sense to have the scale to 150% methylation?

Fig. 7a: Better to show that the loci shown are bivalent in the figure color legend.

Reviewer #2:

Remarks to the Author:

In this revised version the authors have solved several of the initial concerns, such as the presented correlation analysis and new analysis Figure 1 and 2. The addition of the experiments in SUSD2+ cells also improved their results. The authors have now used the same definition for hypermethylation (10%) across the manuscript, which makes more sense.

They have also removed the links to Nanog as they have not a direct mechanism connecting it, as the authors indicated in the rebuttal, it was by "exclusion"....which is very weak.

However, I believe that there still several aspects that remain not answered correctly and therefore making some of the conclusions weak. For example, the leakiness of the system has not been tested and the given explanation does not address the potential problem.

More importantly, I still believe that the main message "the mechanistic link between naïve and cancer" is not well supported. There are few examples displayed, the differences in methylation in these regions are small. Also, the proportion of these overlapping regions is not clear.

Supplementary Figure 9 shows some of these proportions, but they are related to "random" which is not optimal. If this link is going to be made, the authors should show clearly (using the same definition for hypermethylation in both systems) the number of these regions in the Naïve resetting and in all these different cancers. Then a p-value (expected vs unexpected) can be calculated for the common regions and the non-common regions.

In my opinion, although the new analysis on H3K27me3 is interesting, is not enough to strengthen the link that the authors claim.

Finally, as mentioned before the authors should considered to change the word "reprogramming". The given explanation is not satisfactory; indeed it increased my concern. As the authors pointed out, there are now reports of "reprogramming" somatic cells into naïve pluripotency and for the entire field the use of the term reprogramming indicates that. Meanwhile, resetting or togelling indicate Primed to naïve. This will mislead the readers. The authors stated that these regions are common to all naïve states regardless of how they are created. However, they have not done their analysis on all these different naïve cells. Furthermore, if this were the case, they should not use the term reprogramming at all, and just use "naïve condition" since as they claim, they are not studying the specific regions that change between primed and naïve but hypermethylated regions common to any naïve state (which they will need to prove).

Is the authors insist on using the term reprogramming, they should add to the tittle "Reprogramming from Primed to Naive pluripotency..." so there is not confusion.

We are pleased to know that both reviewers appreciated our effort to address many of the concerns raised in response to the first submission. In the attached resubmission, we have addressed the remaining concerns and have highlighted these changes in the edited manuscript.

Reviewer #1 (Remarks to the Author):

The authors have addressed various questions raised in response to the 1st submission. Some of the remaining concerns are listed below which can be addressed through discussions in the paper and figure edits where required. Barring these changes, the paper looks good for publication and brings forth novel findings and concepts.

1) The studies with DNMT3A/B KO and HOX gene 5mc/hmc studies lead to inconclusive results regarding the functional impact of hypermethylation. This is suggested in the rebuttal letter and could be discussed in the paper for the sake of proper context for interpreting and clarity to the reader.

We agree that some extra discussion would aid the interpretation of the data for readers and have now added some further detail to the discussion of the manuscript with regards to the DNMT3A/B double KD and the HOX gene studies and the functional impact of hypermethylation.

2) The dynamic nature of the methylation, i.e. hypermethylation in late-transition followed by demethylation back to almost the primed-state levels should be enhanced further in the discussion/stated in the abstract. Because the current text conveys the message that the “reprogramming” ends up in a final hypermethylated state which is not the case. An accurate description in the abstract should read: TRANSITION to “naïve pluripotency and oncogenic transformation share common...” (Abstract). This is indeed an interesting finding in this model although the functions are not easily discernible.

We agree that the dynamic nature of the methylation is an interesting finding in this model and that this is more aptly described using the phrase “transition to naïve pluripotency”. We have amended the title of the manuscript and the abstract to better convey this message and have added to the discussion to further highlight the dynamic nature of methylation.

3) Some figures regarding KD studies show only primed and late but not early, naive, and various combinations thereof (example Suppl. Fig. 5). Since it is a progression model, it is more useful to know the impact of the manipulation throughout the progression stages. For example, in Suppl. 5b (“qRT-PCR for the transcripts of the DNMT3 family in control and knock-down cell lines, across the period of resetting.”), the data only shows primed and early and not “across the period of resetting”. Important to carefully present this because in Fig. 3a and b “DNMT3A is not solely accountable for all the hypermethylation present”. Was KD efficient throughout, what are the protein level changes in the KDs, etc. are then important considerations which the above mentioned representation of the data could capture. This leads to the important concern that can DNMT3 be attributed to all the methylation changes as suggested! It being the de novo catalytic enzyme, should/could be a key component. However one cannot exclude other factors. Since DNMT1 is also known to possess de novo activity, one cannot completely rule out DNMT1.

Further in Fig. 3b, late transition cells have similar hypermethylation in the DNMT3A and DNMT3B KD in contrast to the conclusion that “DNMT3B knockdown cells fail to hypermethylate to the same extent as control cells and DNMT3A knockdown cells show only a partial reduction in methylation”. And the lack of hypermethylation in the DNMT3A and DNMT3B KD is confined to a set of probes that have 0.2 to 0.4 beta-values, while the sites that have gained high methylation tend to remain hyper

hypermethylated even in the KDs.

The bottom line is the mechanistic links are not so clear, and it is still misleading to state that “We find that de novo methylation is dependent on DNMT3A” (abstract) given the uncertainty. The rest of the paper (the chromatin state links, overlap with cancer methylation, “a putative role of the de novo methyltransferases in stabilization of the naïve pluripotent state” etc.) stands firm without the mechanistic links. Inclusion of mechanism then should be discussed with the proper caveats. This reviewer agrees that it is not trivial to get at the mechanisms by KD/KO experiments because it affects viability/differentiation potential. etc. which are important components of the reprogramming model. Discussing the caveats will provide a proper perspective about the complexity.

We thank the reviewer for pointing out some gaps in the supplementary data. We have now added qPCR data measuring DNMT3A and DNMT3B KD for the late transition of reprogramming to Supplementary Figure 5b to demonstrate that the KD remained efficient throughout the transition to the naïve state. The conclusions regarding DNMT3A and DNMT3B knockdowns during the late transition have been amended in the text to better explain the results as suggested by the reviewer. Moreover, as the effect of DNMT3A and DNMT3B KD is different at different stages of the resetting process, we have removed the statement that “de novo methylation is dependent on DNMT3A” from the abstract so as not to give a misleading message. We have also expanded the discussion to include some of the caveats of the mechanism which should help to provide better perspective to the reader about the complexity of the system, as well as highlighting alternative mechanisms such as the putative *de novo* activity of DNMT1, which cannot be ruled out.

4) KO of both 3A1 and 3A2 would have been a critical experiment at least to resolve the mechanistic basis of the role of the isoforms of 3A (as suggested by R2). KO of 3A1 but not 3A2 is attributed to lack of commercial shRNA targeting 3A2. Does this lack of available shRNA mean inability to design good shRNA? Whatever the case, this can be mentioned in methods.

We have now added to the methods section a brief description of why this experiment was not carried out, due to the inability to design an effective shRNA against the single unique exon of DNMT3A2.

Minor:

Fig. 6c middle panel y-axis scale: does it make sense to have the scale to 150% methylation?

We thank the reviewer for pointing out this error. We have now edited the scale to have a maximum of 100%

Fig. 7a: Better to show that the loci shown are bivalent in the figure colour legend.

We thank the reviewer for this suggestion. We have now edited the figure colour legend to show that both sets of loci are bivalent for Figure 7a and for Supplementary Figure 10.

Reviewer #2 (Remarks to the Author):

In this revised version the authors have solved several of the initial concerns, such as the presented

correlation analysis and new analysis Figure 1 and 2. The addition of the experiments in SUSD2+ cells also improved their results. The authors have now used the same definition for hypermethylation (10%) across the manuscript, which makes more sense.

They have also removed the links to Nanog as they have not a direct mechanism connecting it, as the authors indicated in the rebuttal, it was by “exclusion”which is very weak.

However, I believe that there still several aspects that remain not answered correctly and therefore making some of the conclusions weak. For example, the leakiness of the system has not been tested and the given explanation does not address the potential problem.

With regards to the potential leakiness of the system, it is indeed relevant for the final naïve population but is not relevant for the transition phase where we in fact need robust expression of the construct. Importantly, the primed hESCs we use in the study already contain the NANOG/KLF2 transgenes, so any concerns in the final naïve cells of leaky expression of the transgenes would be an equal concern in the primed cells. This is not the case as evidently, we see a clear difference in methylation levels between the primed and late transition/naïve hESCs, indicating that hypermethylation is not an artefact of the leakiness of the system (i.e should be equally leaky in primed cells) but is a feature of the resetting process. To address this point in a different way, if hypermethylation is a feature of the human naïve cells in general (including the ICM cells), they should overlap independent of the comparison to the NANOG/KLF2 system. We have done a pairwise overlap analysis and show that there is a strong, significant overlap of hypermethylated sites between transgene-independent resetting systems and ICM cells. This new data is shown in the Supplementary Fig 3g.

Regarding the presence of DNA hypermethylation in the final naïve cells, albeit lower than during the transition phase, this is further supported by the data in Supplementary Figure 3 which demonstrates that there is a significant overlap between hypermethylation in the naïve cells in this system and hypermethylation in naïve cells generated through alternative transgene-independent methods of generating naïve cells, including naïve cells *in vivo* and naïve cells isolated directly in naïve media without transitioning through the primed state. Therefore, this overlap cannot be attributed to the potential leakiness in our experimental system. Furthermore, the focus of our mechanistic investigation is during the early transition of the reprogramming process, at which stage doxycycline is actively added to the cells to activate the transgenes.

More importantly, I still believe that the main message “the mechanistic link between naïve and cancer” is not well supported. There are few examples displayed, the differences in methylation in these regions are small. Also, the proportion of these overlapping regions is not clear. Supplementary Figure 9 shows some of these proportions, but they are related to “random” which is not optimal. If this link is going to be made, the authors should show clearly (using the same definition for hypermethylation in both systems) the number of these regions in the Naïve resetting and in all these different cancers. Then a p-value (expected vs unexpected) can be calculated for the common regions and the non-common regions.

In my opinion, although the new analysis on H3K27me3 is interesting, is not enough to strengthen the link that the authors claim.

We respectfully disagree with the reviewer. Our initial results demonstrated that the same genomic regions gain methylation in the transition to naïve pluripotency and cancer. Our analysis of H3K27me3 in normal tissues shows that these regions have similar chromatin states in primed embryonic stem cells and the tissues of origin for different cancers. This suggests that a similar mechanism could be responsible for the hypermethylation in both contexts.

We have now provided more detail as to the degree of overlap between bivalent regions significantly hypermethylated during the transition to naïve pluripotency and significantly hypermethylated in 9 cancer types compared to their corresponding matched normal tissues using Fisher's exact test against the expected background (p -value $< 2 \times 10^{-16}$, percentage fold-change 11.13 testing for all cancers combined). Details of this analysis have been added to the text and provided as a new addition to Figure 7 (Figure 7b). We think this analysis is very useful for our study and the research community, we appreciate this suggestion from the reviewer.

Regarding Supplementary Fig 9 and the use of "random" sites, the analyses tested against random background sets were conducted using the package `regionR` which uses a circular random permutation approach to robustly test genomic enrichments while preserving the non-random distribution of the genome (Gel et al, PMID: 26424858). Importantly, the number and the size of the random regions matches the hypermethylated regions analysed. The random sites were restricted to the EPIC sites using the `resampleRegions` function. We have to do this as the EPIC array is biased towards CpG islands and so you may expect a potential overlap with cancer just by chance - especially considering that cancer relevant CpGs informed a lot of the probes that were chosen. The fact that we see correspondence between gains of DNA methylation in cancer and the transition to naïve pluripotency using different approaches strengthens our conclusions.

Finally, as mentioned before the authors should be considered to change the word "reprogramming". The given explanation is not satisfactory; indeed it increased my concern. As the authors pointed out, there are now reports of "reprogramming" somatic cells into naïve pluripotency and for the entire field the use of the term reprogramming indicates that. Meanwhile, resetting or toggling indicate Primed to naïve. This will mislead the readers. The authors stated that these regions are common to all naïve states regardless of how they are created. However, they have not done their analysis on all these different naïve cells. Furthermore, if this were the case, they should not use the term reprogramming at all, and just use "naïve condition" since as they claim, they are not studying the specific regions that change between primed and naïve but hypermethylated regions common to any naïve state (which they will need to prove).

If the authors insist on using the term reprogramming, they should add to the title "Reprogramming from Primed to Naïve pluripotency..." so there is not confusion.

We understand that the term reprogramming may cause some confusion, so we have now modified the title to read "Transition to naïve pluripotency" rather than reprogramming, to clarify that we are not referring to somatic reprogramming. We have also amended the manuscript to use the word "resetting" in place of "reprogramming" so as to avoid any confusion.

Reviewers' Comments:

Reviewer #1:

Remarks to the Author:

The authors have satisfactorily addressed all concerns. The manuscript looks good and addresses the critical caveats of the model and mechanisms.

Reviewer #2:

Remarks to the Author:

The authors have resolved almost all the remaining questions and I appreciate the changes in the text, I believe that improves clarity.

Although, I respectfully still disagree with them, as they disagree with me, in regards to the mechanistic link between naïve and cancer, I believe that ultimately this is their manuscript, therefore if they are confident of that link they should go ahead with it. That's how science advance, creating discussions and rectifying concepts if necessary eventually.

The overall work is of good quality, they have addressed all other concerns, thus I support the publication of the manuscript.

REVIEWERS' COMMENTS:

Reviewer #1 (Remarks to the Author):

The authors have satisfactorily addressed all concerns. The manuscript looks good and addresses the critical caveats of the model and mechanisms.

Reviewer #2 (Remarks to the Author):

The authors have resolved almost all the remaining questions and I appreciate the changes in the text, I believe that improves clarity. Although, I respectfully still disagree with them, as they disagree with me, in regards to the mechanistic link between naïve and cancer, I believe that ultimately this is their manuscript, therefore if they are confident of that link they should go ahead with it. That's how science advance, creating discussions and rectifying concepts if necessary eventually.

The overall work is of good quality, they have addressed all other concerns, thus I support the publication of the manuscript. We thank both reviewers for their constructive criticism throughout the review process and for their support of publication of the manuscript.

We respect the concerns of Reviewer #2, and agree with their assertion that getting these diverse concepts in front of a wide scientific audience to be tested is the way in which science advances.